# Structural basis of TRAPPIII-mediated Rab1 activation

Aaron MN Joiner[1] ID, Ben P Phillips[2] ID, Kumar Yugandhar[3] ID, Ethan J Sanford[1] ID, Marcus B Smolka[1] ID, Haiyuan Yu[3] ID, Elizabeth A Miller[2] ID & J Christopher Fromme[1,*] ID

## Abstract

The GTPase Rab1 is a master regulator of the early secretory pathway and is critical for autophagy. Rab1 activation is controlled by its guanine nucleotide exchange factor, the multisubunit TRAPPIII complex. Here, we report the 3.7 Å cryo-EM structure of the *Saccharomyces cerevisiae* TRAPPIII complex bound to its substrate Rab1/Ypt1. The structure reveals the binding site for the Rab1/Ypt1 hypervariable domain, leading to a model for how the complex interacts with membranes during the activation reaction. We determined that stable membrane binding by the TRAPPIII complex is required for robust activation of Rab1/Ypt1 *in vitro* and *in vivo*, and is mediated by a conserved amphipathic α-helix within the regulatory Trs85 subunit. Our results show that the Trs85 subunit serves as a membrane anchor, via its amphipathic helix, for the entire TRAPPIII complex. These findings provide a structural understanding of Rab activation on organelle and vesicle membranes.

**Keywords** autophagy; GTPase; guanine nucleotide exchange factor; membrane trafficking; Rab1
**Subject Categories** Membranes & Trafficking; Structural Biology
**The EMBO Journal (2021) 40: e107607**
See also: **A Galindo *et al*** (June 2021) and **BS Glick** (June 2021)

## Introduction

The GTPase Rab1 is a master regulator of both the early secretory pathway and autophagy (Jedd *et al*, 1995; Lynch-Day *et al*, 2010; Zoppino *et al*, 2010; Hutagalung & Novick, 2011). In budding yeast, the Rab1 protein is encoded by the *YPT1* gene (Segev *et al*, 1988; Segev, 1991). Once activated by GTP binding, Rab1/Ypt1 recruits vesicle tethering factors such as Uso1, Atg11, and the COG complex to initiate membrane fusion events (Jedd *et al*, 1995; Yamakawa *et al*, 1996; Cao *et al*, 1998; Suvorova *et al*, 2002; Lipatova *et al*, 2012). Proper localization of Rab1/Ypt1 is directly linked to its activation state, and due to a slow intrinsic rate of exchange, the key regulatory step of Rab1/Ypt1 activation is catalyzed by its GEF (guanine nucleotide exchange factor), the multisubunit TRAPPIII complex.

In the late secretory pathway, TRAPPII (a distinct, but related multisubunit complex) activates the small GTPase Rab11 (Jedd *et al*, 1995; Jones *et al*, 2000; Pinar *et al*, 2015; Thomas & Fromme, 2016; Thomas *et al*, 2019). While the TRAPPII and TRAPPIII complexes share a core set of subunits, the distinguishing feature of the TRAPPIII complex is the presence of the Trs85 subunit (TRAPPC8 in metazoans) (Sacher *et al*, 1998, 2000, 2001, 2019; Lynch-Day *et al*, 2010). The metazoan and Aspergillus TRAPPIII complexes possess three additional subunits that are not present in budding yeast (Bassik *et al*, 2013; Riedel *et al*, 2018; Pinar *et al*, 2019; Pinar & Peñalva, 2020) (Appendix Table S1), but the 7 genes encoding the 8 subunits of the budding yeast TRAPPIII complex are highly homologous with their metazoan paralogs (Koumandou *et al*, 2007; Sacher *et al*, 2019).

Loss of Trs85 results in a major reduction of Rab1/Ypt1 activation *in vivo* and *in vitro*, and causes defects in autophagy and aberrant secretion. TRAPPIII has been reported to function at many cellular locations important for secretion and autophagy, including COPII vesicles, the Golgi complex, Atg9 vesicles, and the pre-autophagosomal structure in both budding yeast and metazoans (Meiling-Wesse *et al*, 2005; Nazarko *et al*, 2005; Lynch-Day *et al*, 2010; Kakuta *et al*, 2012; Lipatova *et al*, 2012; Bassik *et al*, 2013; Tan *et al*, 2013; Imai *et al*, 2016; Lamb *et al*, 2016; Zhao *et al*, 2017; Thomas *et al*, 2018; Stanga *et al*, 2019; Zhang *et al*, 2020).

Previous reports have presented crystal structures of the TRAPP core subunits and negative-stain electron microscopy analysis of the TRAPPIII complex, which provided an important foundation for understanding the overall architecture and function of the entire TRAPPIII complex (Jang *et al*, 2002; Kim *et al*, 2005, 2006; Tan *et al*, 2013). The nucleotide exchange mechanism was also elucidated using a structure of the core subunits bound to Rab1/Ypt1, and the active site comprises portions of four subunits (Bet5, Bet3, Trs31, and Trs23) (Kim *et al*, 2006; Cai *et al*, 2008). However, the mechanistic role of Trs85 in the TRAPPIII complex has remained unclear, the structure of Trs85 (and the entire TRAPPIII complex) has not been determined, and how the complex interacts with membranes in order to activate Rab1/Ypt1 at an organelle membrane surface is unknown.

---

1 Department of Molecular Biology and Genetics/Weill Institute for Cell and Molecular Biology, Cornell University, Ithaca, NY, USA
2 MRC Laboratory of Molecular Biology, Cambridge, UK
3 Department of Computational Biology/Weill Institute for Cell and Molecular Biology, Cornell University, Ithaca, NY, USA
*Corresponding author. Tel: +1 607 255 1016; E-mail: jcf14@cornell.edu

In order to define the role of the Trs85 subunit within the TRAP-PIII complex, we have used a comprehensive approach combining cryo-EM, cross-linking mass spectrometry, biochemical reconstitution of membrane binding and nucleotide exchange activity, and *in vivo* functional analysis. Here, we report the high-resolution structure of the intact yeast TRAPPIII complex bound to its substrate GTPase Rab1/Ypt1, representing the key intermediate of the nucleotide exchange reaction. The structure reveals that Trs85 possesses both an N-terminal GTPase-like fold and a C-terminal α-solenoid motif. The α-solenoid connects Trs85 to the TRAPP core, interacting with a core residue mutated in the human disease SEDT. A portion of the flexible "HVD" region of Rab1/Ypt1, which links the GTPase to the membrane, is visible in the structure bound to the Trs31 core subunit. The positioning of the HVD and the electrostatics of the complex suggest a specific orientation on the membrane surface. We identify a conserved amphipathic α-helix motif on the putative

membrane-binding surface of Trs85 and determine that this motif is required for stable membrane binding and Rab1/Ypt1 activation both *in vitro* and *in vivo*. Our results indicate that this region is also required for the function of Trs85 in autophagy, signifying that TRAPPIII uses the same membrane-binding mechanism to activate Rab1/Ypt1 for both secretion and autophagy.

# Results

### Trs85 anchors the TRAPPIII complex to the membrane surface

We previously reported that the Trs85 subunit is required for robust Rab1/Ypt1 activation at the Golgi complex *in vivo* and on synthetic liposomes *in vitro* (Thomas *et al*, 2018) (Fig 1A and B). In contrast, Trs85 was dispensable for GEF activity in the absence of membranes

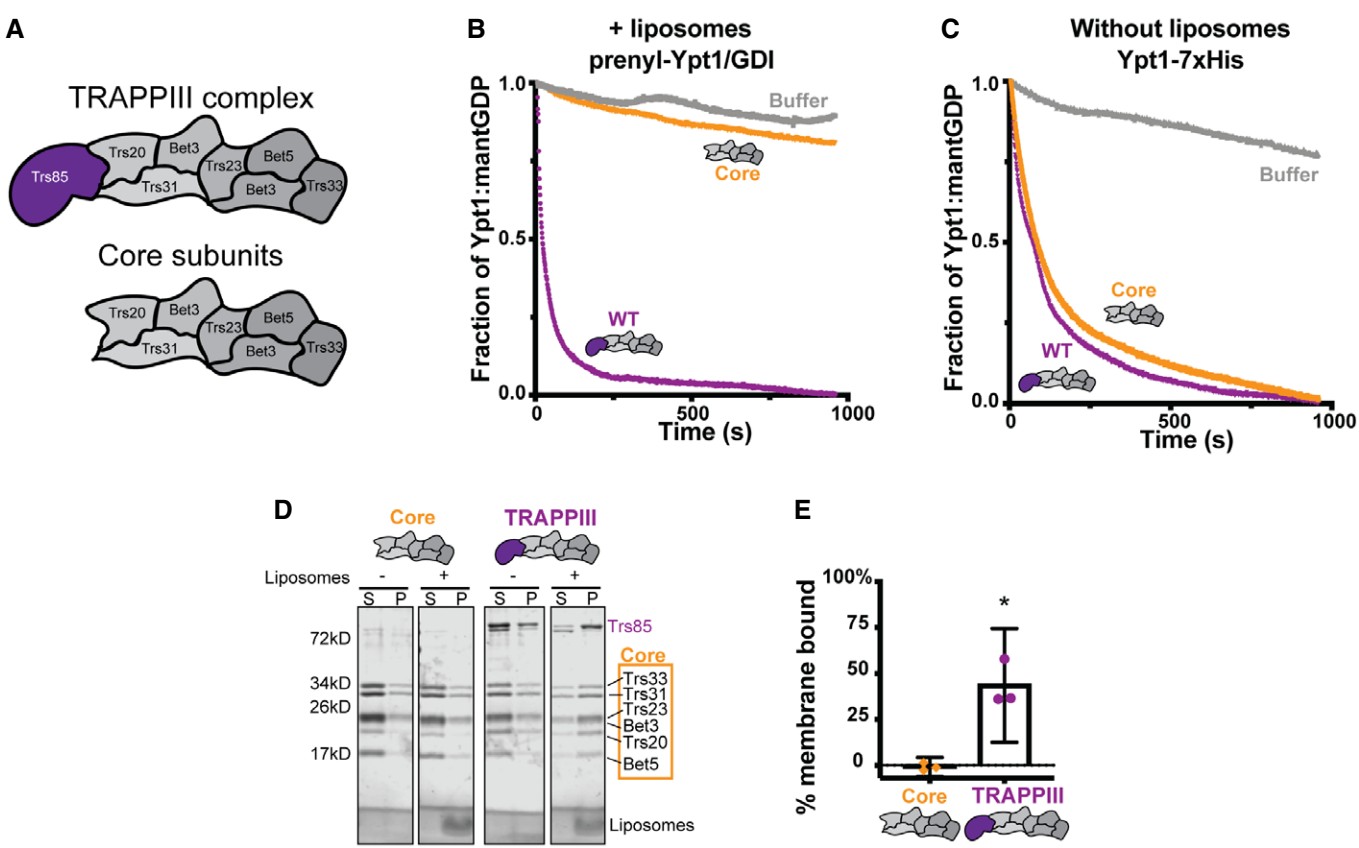

**Figure 1. The Trs85 subunit anchors TRAPPIII to membranes.**

A  Cartoon depicting the TRAPPIII complex subunits and the TRAPP core subunits. Trs85 (purple) is the TRAPPIII-specific subunit.

B  *In vitro* nucleotide exchange assay of prenylated-Rab1/Ypt1-GDI substrate in the presence of synthetic liposomes. The TRAPPIII complex (purple) is significantly more active than the TRAPP core alone (yellow). Representative of $n = 3$ independent experiments.

C  Assays performed as in (B), but using a non-prenylated Rab1/Ypt1 substrate and in the absence of synthetic liposomes. Exchange activity with and without the Trs85 subunit are equivalent. Representative of $n = 3$ independent experiments.

D  *In vitro* membrane pelleting assay comparing the TRAPPIII complex versus the core subunits. While both complexes exhibit a small amount of pelleting in the absence of synthetic liposomes, in their presence only the TRAPPIII complex is enriched in the membrane-bound fraction (S = supernatant, P = pellet) Representative of $n = 3$ independent experiments.

E  Quantification of (D). Average % membrane binding between three assays is plotted, and the error bars represent the 95% confidence intervals. The value for each replicate is depicted. The negative values arise from subtraction of the average amount of protein that pellets in the absence of liposomes. Analyzed by unpaired, two-tailed Student's *t*-test with Welch's correction, *$P = 0.0229$. $n = 3$ independent experiments.

Source data are available online for this figure.

(Thomas *et al,* 2018) (Fig 1C). We hypothesized that a primary function of Trs85 might be to directly interact with the membrane surface. Although we were unable to purify the isolated Trs85 subunit, we compared membrane binding of the purified TRAPPIII complex with and without the Trs85 subunit, using an *in vitro* liposome pelleting assay (Fig 1D). Consistent with our hypothesis, the intact TRAPPIII complex bound to synthetic liposomes and the Trs85 subunit was required for stable membrane association (Fig 1E).

**Cryo-EM structure of the yeast TRAPPIII-Rab1/Ypt1 complex**

To establish the mechanism of Trs85 function within the complex, we sought to resolve the high-resolution structure of the intact budding yeast (*S. cerevisiae*) TRAPPIII complex. Furthermore, the published crystal structure of TRAPP core subunits bound to the Rab1/Ypt1 substrate lacks the critical "HVD" (hypervariable domain) of Rab1/Ypt1 (Cai *et al,* 2008), which has been shown to play an important role in substrate recognition by the TRAPP complexes (Thomas *et al,* 2019). We therefore assembled a stable complex in which TRAPPIII was bound to full-length yeast Rab1/Ypt1 in order to also visualize any interactions involving the HVD (Fig EV1A). For this purpose, we used a yeast Rab1/Ypt1 construct in which the two C-terminal cysteine residues that are prenylated *in vivo* were replaced with an affinity tag for purification.

This complex was well behaved and suitable for high-resolution cryo-EM experiments (Fig EV2A and B), but the complex exhibited a strongly preferred orientation on frozen grids. Initial datasets indicated that approximately 95% of particles adopted a single orientation and the other 5% adopted an orientation that was only slightly different from the primary orientation. To capture additional views, we therefore imaged grids tilted by 30°. Although this resulted in a marked improvement in the resulting reconstruction, significant orientation bias remained that precluded confident *de novo* building of Trs85. We therefore collected additional data using grids tilted at 40° and 45°. Although these data were of lower quality due to the higher tilt angle, combination of data from the three tilt angles provided sufficient orientation coverage to generate a 3.7 Å reconstruction suitable for *de novo* model building from 69,315 particles (Figs 2A and B, EV2C–G and EV3, and Table 1). For the core subunits and GTPase substrate, we docked and rebuilt available crystal structures of subunits from yeast and other species (Kim *et al,* 2005, 2006; Cai *et al,* 2008). For Trs85, we used trRosetta (Yang *et al,* 2020) to generate hypothetical models of Trs85 from budding yeast and several other organisms. The overall topologies of these prediction models were quite similar and matched the 3D-reconstruction reasonably well. The trRosetta

model of budding yeast Trs85 was therefore used as a guide for *de novo* building of the Trs85 model.

The TRAPPIII complex adopts a narrow rod-like structure (Fig 2A and B), in agreement with the published negative-stain EM structure (Tan *et al,* 2013). The structures of each of the core subunits and the GTPase substrate are largely similar to the published crystal structures (Fig EV4A–G). Some small differences were noted; for example, we observed covalently bound palmitate molecules in both Bet3 subunits, whereas only one of the Bet3 subunits appeared to be bound to palmitate in the previous structure of the catalytic core (Cai *et al,* 2008). These palmitates are thought to stabilize the folding of the Bet3 subunits, rather than mediate interactions with the lipid membrane (Turnbull *et al,* 2005; Kümmel *et al,* 2006). The interaction between the TRAPP active site and the globular domain of Rab1/Ypt1 is essentially the same as that observed in the crystal structure previously determined by the Reinisch group (Cai *et al,* 2008). However, we observed a novel interaction with the Rab1/Ypt1 HVD that we discuss further below.

To validate the final structural model, we utilized cross-linking mass spectrometry (XL-MS) to identify lysine residues in proximity to each other (Figs 2C and D, and EV1B, Table EV1). From 108 total cross-links, 32 could be mapped to residues modeled in the structure (Fig 2E). 31 of these 32 cross-links were within the maximum distance constraint for disuccinimidyl sulfoxide (DSSO), 30 Å, with most occurring between residues < 20 Å apart (Fig 2F). The other cross-link (between Lys78 of Trs31 and Lys85 of Bet3) spanned a distance of 45 Å and is likely a false positive because those two cross-linked residues were also far apart in the crystal structure of these subunits bound to Rab1/Ypt1 (Cai *et al,* 2008). However, we cannot rule out the possibility that this cross-link might have captured some of the dynamic movement of the complex in solution. The remaining 76 cross-links could not be mapped onto the model because they involved one or more residues located in unmodeled loops (Table EV1). For each of these residues, we confirmed that the position of these unstructured loops in the model was consistent with the existence of a cross-link. Overall, the XL-MS results contribute support for the atomic model of the complex.

**The C-terminus of Trs85 binds to the core subunits Trs20 and Trs31**

The cryo-EM structure reveals that the N-terminal portion of Trs85 adopts a GTPase-like fold and the C-terminal portion folds into a twisting α-solenoid comprising five pairs of α-helices (Fig 3A). The α-solenoid connects Trs85 to the TRAPP core through a surface that

---

**Figure 2.   Cryo-EM structure of the TRAPPIII-Rab1/Ypt1 nucleotide exchange complex.**

A   Cryo-EM density map colored by each subunit. Bottom panel is rotated 90° relative to the top panel.
B   Colored as in (A), the refined atomic model of the TRAPPIII complex.
C   Network map depicting all of the 108 cross-links found between the TRAPPIII complex subunits and Rab1/Ypt1. Blue lines are indicative of inter-protein cross-links, and red lines denote cross-links detected within a single protein. The number of residues in each peptide is labeled within each box, and the relative position of the cross-link represents its location within the primary sequence.
D   The fraction of total cross-linking sites observed in each polypeptide.
E   The cross-links mapped onto the refined model. Cross-links are shown as black lines connecting black spheres.
F   A histogram depicting the length of each cross-link that was mapped to the structural model. The distance was measured between the α-carbons of each pair of cross-linked residues. Only one of the cross-links is beyond the maximum theoretical restraint (30 Å) of the cross-linking chemical, DSSO.

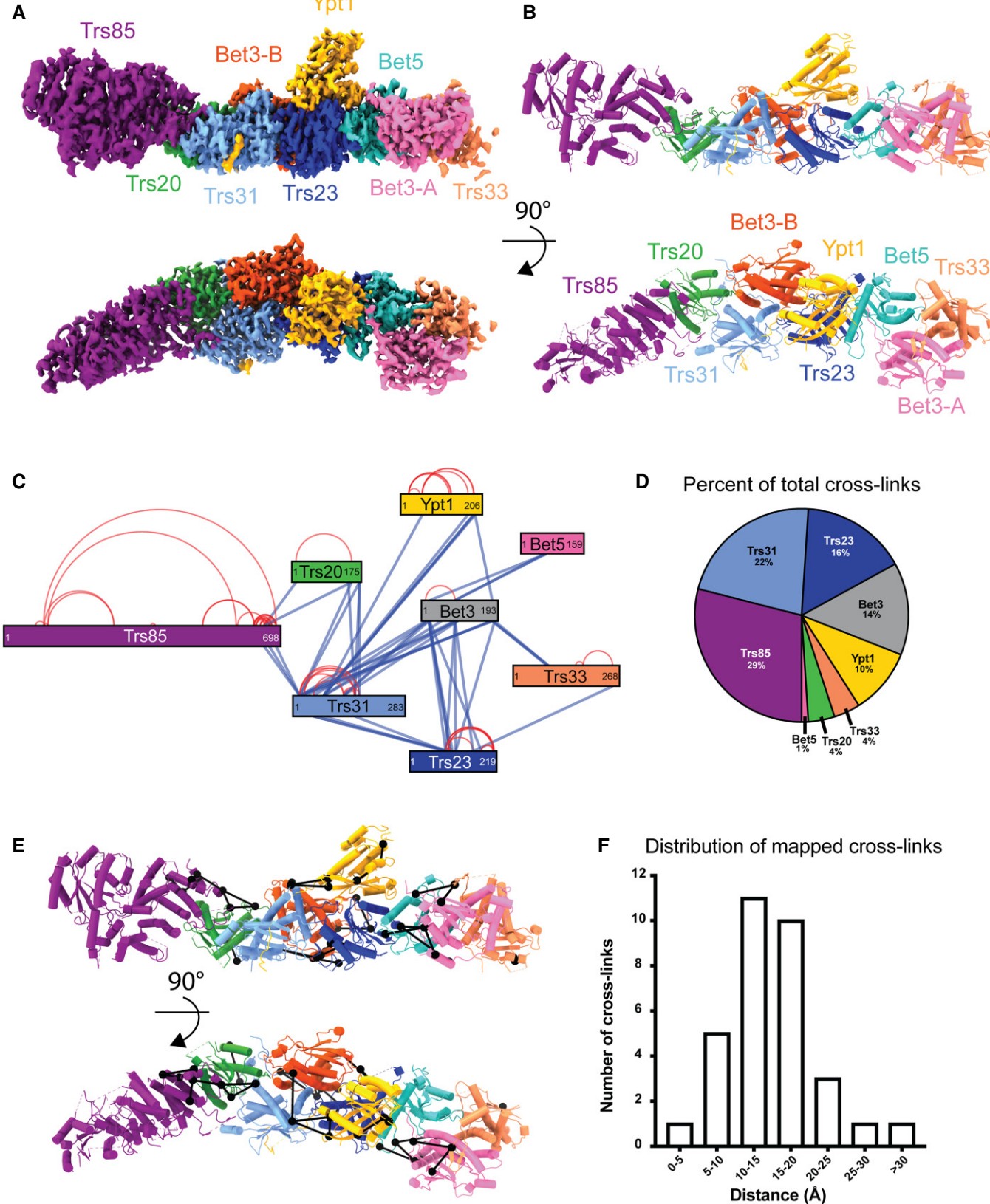

**Figure 2.**

**Table 1. Cryo-EM model validation statistics.**

| Composition (#) | |
|---|---|
| Chains | 10 |
| Atoms | 14,947 |
| Protein residues | 1,838 |
| Ligands | 2 palmitate |
| Bonds (RMSD) | |
| Length (Å) (# > 4σ) | 0.005 (0) |
| Angles (°) (# > 4σ) | 1.122 (1) |
| MolProbity score | 1.81 |
| Clash score | 7.08 |
| Ramachandran plot (%) | |
| Outliers | 0.00 |
| Allowed | 6.26 |
| Favored | 93.74 |
| Rama-Z: Ramachandran plot Z-score (RMSD) | |
| Whole (N = 1,754) | 0.96 (0.20) |
| Helix (N = 842) | 0.37 (0.17) |
| Sheet (N = 231) | 1.61 (0.30) |
| Loop (N = 681) | 1.45 (0.26) |
| Rotamer outliers (%) | 0 |
| Cβ outliers (%) | 0 |
| CaBLAM outliers (%) | 1.25 |
| ADP (B-factors) | |
| Protein (min/max/mean) | 83.62/170.92/118.88 |
| Ligand (min/max/mean) | 98.80/142.33/120.56 |
| FSC Resolution Estimates (Å) (0.143/0.5) | |
| FSC half-map / half-map | 3.7/4.2 |
| FSC model / masked full map | 3.2/3.8 |
| FSC model / masked refinement half-map | 3.2/4.0 |
| FSC model / masked validation half-map | 3.4/4.1 |
| Model versus Data | |
| CC mask | 0.82 |
| CC box | 0.88 |
| CC peaks | 0.76 |
| CC volume | 0.82 |
| CC ligands | 0.72 |
| Other resolution estimates (Å) | |
| $d_{99}$ (unsharpened / sharpened / density-modified) | 4.4/3.4/3.3 |
| $d_{model}$ (unsharpened / sharpened / density-modified) | 3.8/3.6/3.3 |
| $d_{FSC\text{-}model}$ (unsharpened / sharpened / density-modified) | 3.2/3.2/3.0 |

is well conserved among Trs85 homologs throughout eukaryotes (Fig 3B). This interaction surface is described further below. The GTPase-like fold in Trs85 has multiple insertions relative to a typical GTPase protein and does not appear capable of binding nucleotide. Key nucleotide-binding residues are not conserved in Trs85 and an

α-helix formed by residues 46–55 occupies the region where nucleotides bind to GTPases.

The structure indicates that the C-terminal α-solenoid region of Trs85 interacts with the TRAPP core. We determined that the C-terminal 198 amino acids of Trs85 (Trs85[501–698]) are sufficient to maintain this interaction during purification of the recombinant complex (Fig 4A and B), and several cross-links were detected between the C-terminal region of Trs85 and either Trs20 or Trs31 in the XL-MS experiments (Figs 2D and E, and 4D, and Table EV1).

There are two interfaces between Trs85 and the TRAPP core subunits: an extensive interaction between Trs85 and Trs20 (Fig 4E), involving 31 residues and encompassing a total surface area of roughly 2,250 Å, and a much smaller interaction between Trs31 and part of a largely unstructured loop of Trs85. This loop spans residues ~580–612 of Trs85 and appears to be an insertion specific to *S. cerevisiae*.

The interface between Trs85 and Trs20 is quite conserved (Fig 3B) and involves Trs20 Asp46 (Fig 4E), a residue known to be important for assembly of both the TRAPPII and TRAPPIII complexes (Zong *et al*, 2011; Brunet *et al*, 2013; Taussig *et al*, 2014) that is mutated in spondyloepiphyseal dysplasia tarda (SEDT) (Sacher *et al*, 2019). This and nearby negatively charged residues of Trs20: Asp12, Glu49, Asp50, and Asp93, interact with multiple positively charged residues of Trs85: Arg618, Arg619, Arg620, and Lys621 (Fig 4E). To test the importance of this interface *in vivo*, we generated alanine and charge reversal substitutions for the Trs85 residues and assessed their ability to provide Trs85 function. Because loss of Trs85 is synthetically lethal with C-terminal tagging of Bet3 (Sacher *et al*, 2001), we created a sensitized strain background for *trs85Δ* complementation tests in which the core Bet3 subunit is tagged at its C terminus. All of the substitutions dramatically impaired Trs85 function (Fig 4F). Even substitution of just one of these residues, Trs85 Arg620, with either alanine or glutamate, significantly impaired Trs85 function, with R620E exhibiting a stronger phenotype than R620A (Fig 4F).

To determine whether these phenotypes were due to disruption of the Trs85-Trs20 interaction, we investigated the assembly status of the TRAPPIII complex by monitoring the amount of the core subunit Trs23 that co-immunoprecipitated with the Trs85 mutant alleles (Fig 4G). The quadruple substitutions and R620E single substitution mutant essentially abolished the interaction between Trs85 and the core subunits, whereas the R620A substitution mutant exhibited a partial effect. As the severity of complex disruption correlated with the growth phenotypes in the complementation test, these results provide a strong validation of the importance of the Trs85–Trs20 interface observed in the cryo-EM structure (Fig 4F and G).

We previously reported that Trs85 is required for robust activation of Rab1/Ypt1 at the Golgi complex (Thomas *et al*, 2018). We tested the ability of these complex-disrupting mutants to activate Rab1/Ypt1 at the Golgi in otherwise wild-type cells (Fig 4H). As predicted, the quadruple substitution mutants and R620E single substitution were unable to robustly activate Rab1/Ypt1, as monitored by the loss of punctate RFP-Ypt1 localization. The R620A substitution again exhibited an intermediate phenotype. Interestingly, the complex-disrupting substitution mutants also resulted in mislocalization of Trs85 (Fig 4H), suggesting that Trs85 is not able to stably bind to membranes by itself when not incorporated into the TRAPPIII complex.

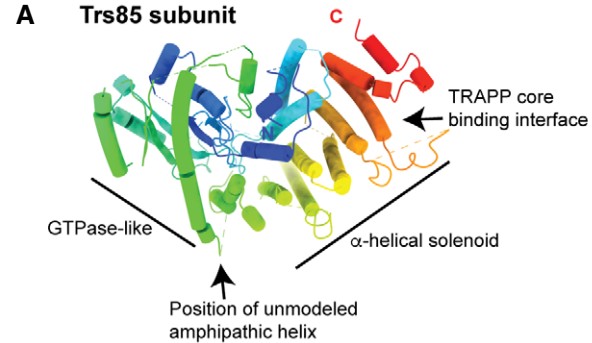

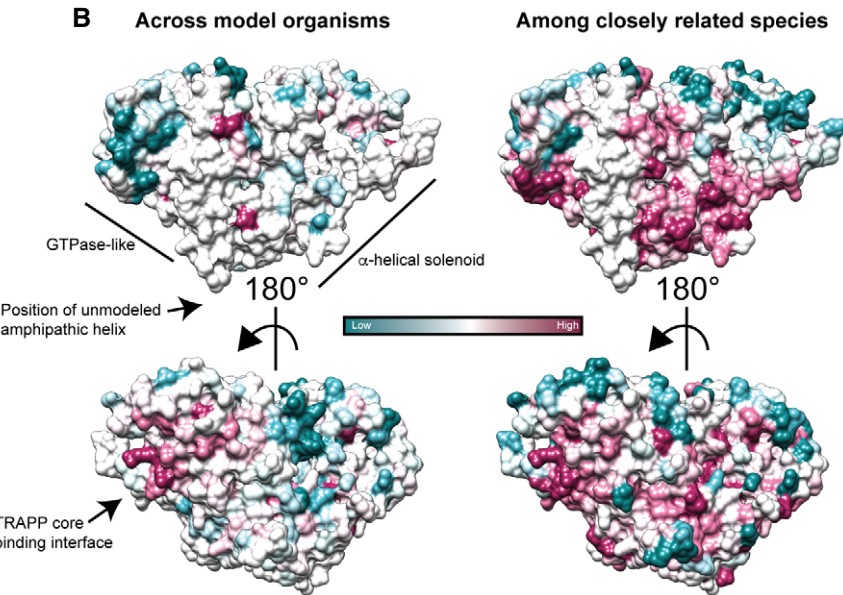

**Figure 3. Architecture and conservation of the Trs85 subunit.**

A The Trs85 subunit colored from N- (blue) to C-terminus (red). Several features are highlighted and described in the text.

B Surface view of Trs85 colored by ConSurf analysis. On the left is the conservation across model organisms and humans, on the right conservation was estimated based on more closely related species. The TRAPP core binding region is highly conserved.

To examine the importance of the less conserved interface between Trs85 and Trs31 (Fig 2B), we tested a mutant lacking a portion of the partially structured loop designed to disrupt this interaction without disrupting the folding of Trs85 (Trs85$^{\Delta575–603}$). This Trs85 mutant exhibited a minor temperature-sensitive growth phenotype and a partial loss of interaction with the TRAPP core in co-IP experiments (Fig 4F and G). To further examine the effect of this mutant, we attempted purification of recombinant TRAPPIII harboring this loop deletion. The complex remained largely intact during nickel-ion affinity purification but then dissociated during gel filtration chromatography (Fig 4C), suggesting a weakened ability of this Trs85 mutant to stably associate with the core subunits. In contrast, we were unable to purify the quadruple substitution mutant complexes disrupting the Trs20 interaction, consistent with an inability to purify the isolated Trs85 subunit. Therefore, the loop comprising residues 575-603 plays a subsidiary role in linking Trs85 to the TRAPP core subunits by interacting with Trs31.

**A model for the orientation of TRAPPIII on the membrane surface**

The TRAPP complexes identify their specific substrates in part through recognition of the GTPase HVD (Thomas *et al*, 2019). It is likely that other Rab-GEFs also recognize the HVD regions of their GTPase substrates (Chavrier *et al*, 1991; Dunn *et al*, 1993; Li *et al*, 2014), but interactions between GEFs and GTPase HVD regions have not been structurally characterized. In fact, HVD regions are often intentionally omitted from structural studies because of their "unstructured" nature (Dong *et al*, 2007; Cai *et al*, 2008). In our cryo-EM reconstruction of the TRAPPIII-GTPase complex, we observed a region of strong but unexpected density on the surface of the complex that we have confidently assigned as corresponding to a portion of the Rab1/Ypt1 HVD (Figs 2A and B, and EV4H). There are multiple pieces of evidence supporting this assertion: (i) At lower threshold levels of the reconstruction, there is a nearly continuous tube of density connecting this region of unexpected density

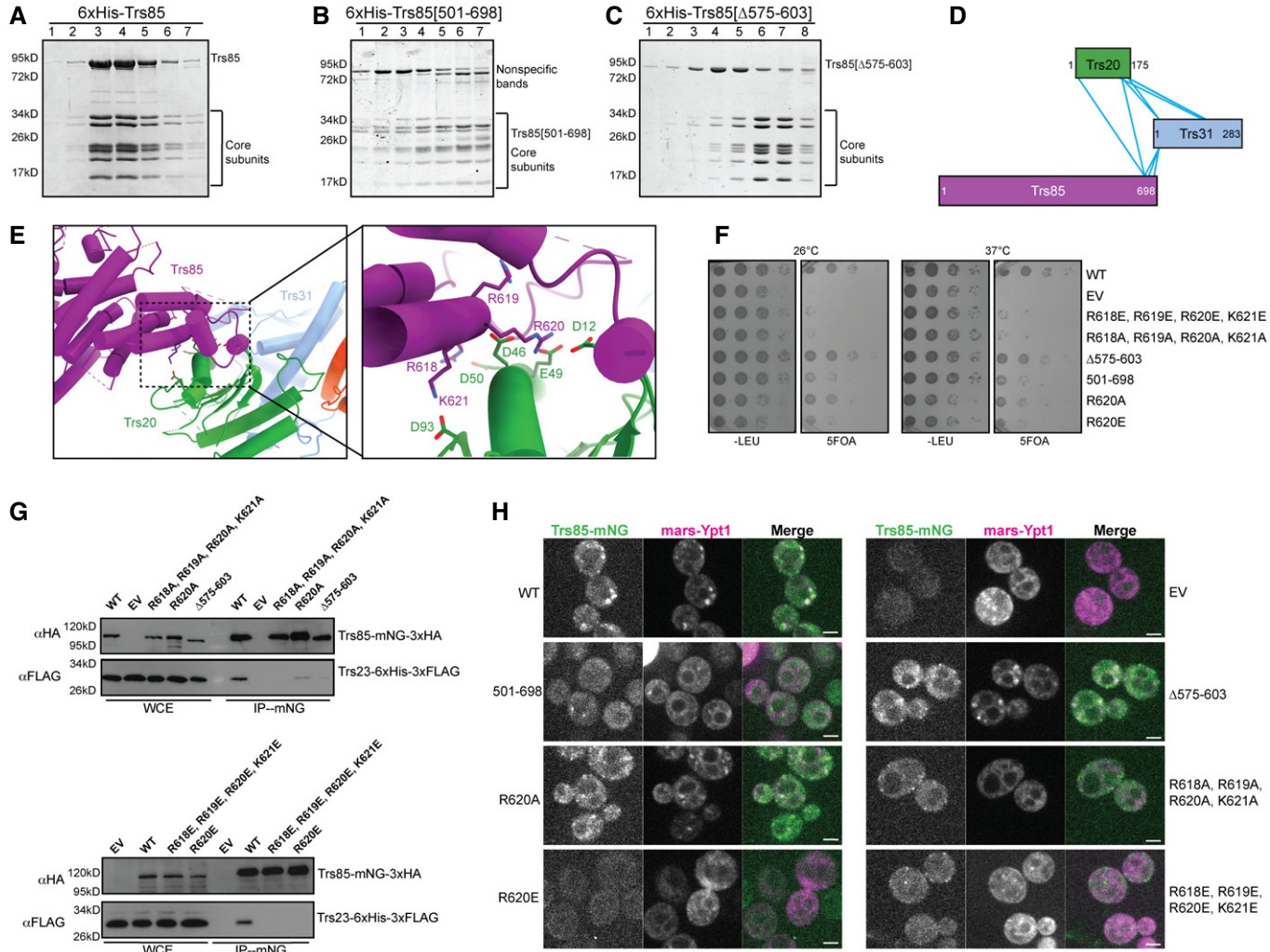

**Figure 4. The C-terminus of Trs85 binds to the TRAPP core.**

A Representative Coomassie-stained gel of size exclusion chromatography fractions after purification of recombinant wild-type TRAPPIII complex. Representative of *n* = 3 independent experiments.

B As in (A), using a Trs85 truncation containing only the final 198 amino acids. NOTE: Trs85[501–698] is ~25 kD in size and now migrates at the same size as some of the core subunits. The species migrating near 85 kD are contaminants present due to the lower expression level of this mutant construct. Representative of *n* = 3 independent experiments.

C As in (A, B) using a Trs85 mutant with a truncation of the loop from 575-603. Trs85[Δ575–603] appears to dissociate from the core during chromatography, indicating a reduced affinity for the core subunits. Representative of *n* = 3 independent experiments.

D Network map showing the relative positions of cross-links between the C-terminus of Trs85 and the core subunits Trs31 and Trs20.

E View of the cryo-EM structural model showing the interaction surface between Trs85, Trs31, and Trs20. Inset depicts several basic residues from Trs85 that are in close proximity to several acidic residues of Trs20.

F Yeast complementation assays performed in a sensitized strain (*BET3-GFP*::HIS3, *trs85Δ*::KANMX) to assess the functionality of Trs85 mutants within the region shown in (E). Representative of *n* = 3 independent experiments. (WT = wild-type, EV = empty vector).

G Coimmunoprecipitations of Trs23-6xHis-3xFLAG with the same Trs85 mutants. Representative of *n* = 3 independent experiments. (WCE = whole-cell extract, WT = wild-type, EV = empty vector).

H Live cell fluorescence microscopy of Trs85-mNeonGreen and mRFPmars-Ypt1. Localization of Rab1/Ypt1 to punctate structures is indicative of its activation by TRAPPIII. Scale bar is 2 μm. Representative of *n* = 3 independent experiments. (WT = wild-type, EV = empty vector).

Source data are available online for this figure.

with the C terminus of the Rab1/Ypt1 nucleotide-binding domain (NBD) (Fig 5A); (ii) in preliminary cryo-EM experiments, we observed 2D classes corresponding to complexes either with or without bound Rab1/Ypt1, and this unexpected density was only present when Rab1/Ypt1 was also bound (Fig 5B); and (iii) XL-MS data identified cross-links between the Rab1/Ypt1 HVD and the Trs31 subunit, occurring in the region between the C-terminus of the Rab1/Ypt1 globular domain and the unexpected density (Fig 5C and D, Table EV1). Therefore, although the unexpected cryo-EM density is not clear enough to unambiguously assign specific

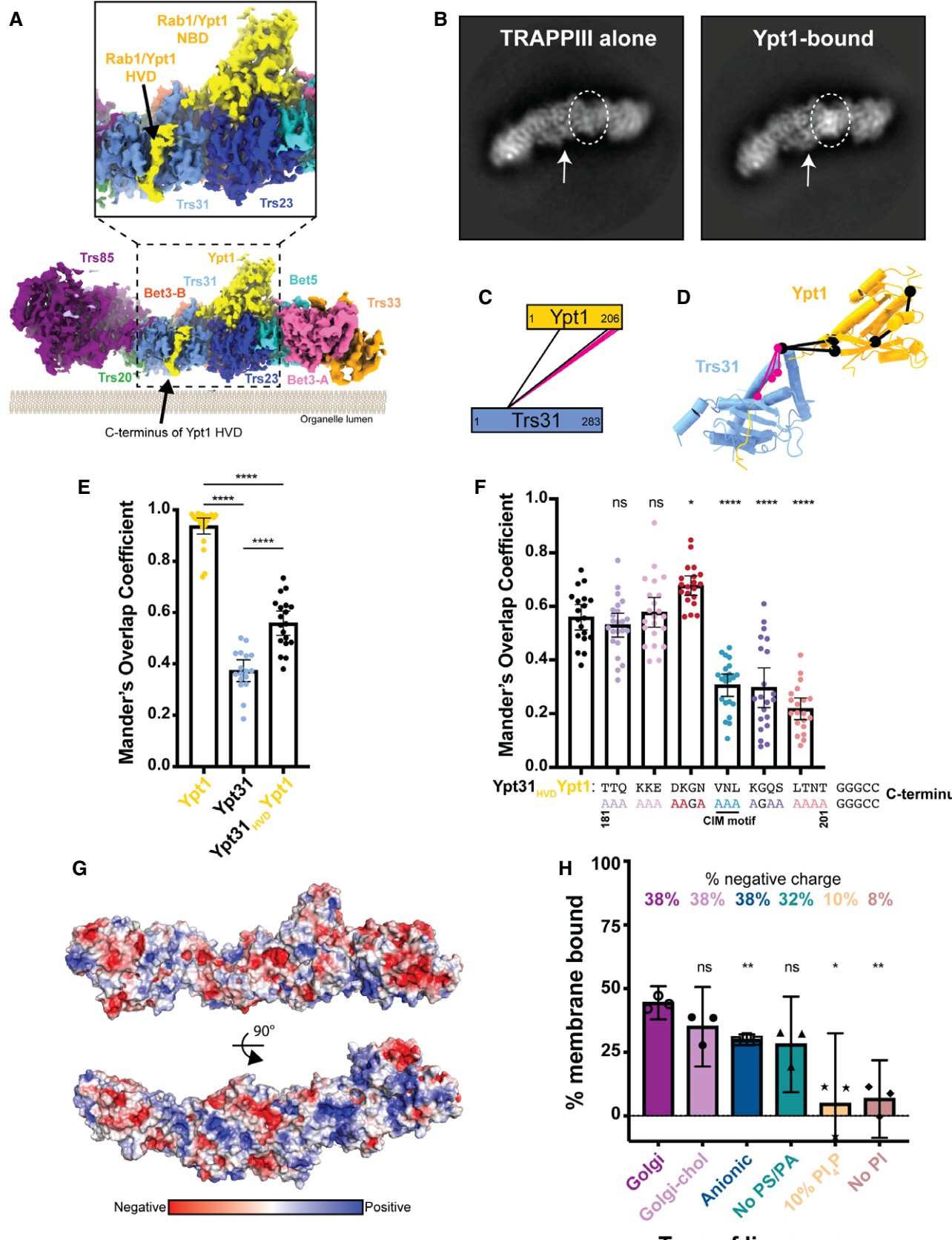

**Figure 5.**

**Figure 5.   The Rab1/Ypt1 HVD points to the membrane-binding surface.**

A   Cryo-EM density is shown at a low threshold to highlight the almost continuous density connecting the Rab1/Ypt1 nucleotide-binding domain (NBD) to its HVD (hypervariable domain). Inset is a closeup of the density corresponding to the Rab1/Ypt1 HVD region. Density is colored by subunit as in Fig 2, Rab1/Ypt1 is yellow. Density is depicted as it is likely oriented at the surface of organelle membranes.

B   2D class averages showing extra density in this region in the presence of Rab1/Ypt1. Circle highlights the Rab1/Ypt1 NBD, and the arrow points to the HVD density.

C   Network diagram of cross-links observed between Trs31 and Rab1/Ypt1. Many cross-links are observed at the C-terminal region of Rab1/Ypt1. Cross-links involving residues present in the atomic model are colored black, while cross-links between unmodeled residues are colored pink.

D   As in (C), but the cross-links are mapped onto the structure. Cross-links involving residues present in the atomic model are colored black, while cross-links between unmodeled residues are colored pink.

E   Quantification of the colocalization of TRAPPIII with mitochondrially anchored Rab substrates. Average Mander's Overlap Coefficient (MOC) is plotted with 95% confidence intervals. Each dot represents the MOC for a cropped image containing 2-9 cells. Samples are compared using an ordinary one-way ANOVA with Tukey's multiple comparisons test, ****$P < 0.0001$.

F   Quantification of the colocalization of TRAPPIII with mitochondrially anchored Rab substrates. The sequence of wild-type Rab1/Ypt1 HVD is in black, and the substituted residues are colored to correspond with their location on the graph. The "CIM" motif is required for enzymatic prenylation of the Rab, but these chimeras are anchored to the plasma membrane independently of prenylation, via Fis1-protein fusion. Average Mander's overlap coefficient (MOC) is plotted with 95% confidence intervals. Each dot represents the MOC for a cropped image containing 2–9 cells. Samples are compared using an ordinary one-way ANOVA with Tukey's multiple comparisons test, but significance indicators are only present for each substitution mutant compared with "wild-type" Ypt31$_{HVD}$Ypt1: versus $_{181}$TTQ $P = 0.9924$, versus $_{184}$KKE $P = 0.9997$, versus $_{187}$DKGN *$P = 0.0145$, versus $_{191}$VNL, $_{194}$KGQS, $_{198}$LTNT ****$P < 0.0001$, ns = not significant.

G   Surface representation of the TRAPPIII-Rab1/Ypt1 complex colored by electrostatic potential. Top panel is oriented as in (A), and the bottom panel is rotated 90° to visualize the most positive surface, which likely lies at the surface of the membrane.

H   Quantification of *in vitro* TRAPPIII complex membrane pelleting assays performed with various lipid compositions. The strongest binding was observed with the most negatively charged liposomes. Binding is compared with wild-type (WT) on "Golgi" liposomes using an unpaired, two-tailed Student's *t*-test with Welch's correction: Golgi versus Golgi-chol $P = 0.1067$, versus Anionic **$P = 0.0063$, versus No PS/PA $P = 0.0522$, versus 10% PI4P *$P = 0.0207$, versus No PI **$P = 0.0035$, ns = not significant. The value for each of $n = 3$ independent experiments is depicted. Error bars represent 95% confidence intervals.

residues of Rab1/Ypt1 to this region, we are confident that it corresponds to some portion of the Rab1/Ypt1 C-terminal HVD.

This interpretation is further supported by experiments using a quantitative *in vivo* anchor-away assay, which has previously been utilized to interrogate GEF-GTPase interactions (Thomas *et al,* 2019). Nucleotide-free RFP-tagged Rab1/Ypt1, when ectopically localized to mitochondria, robustly recruits TRAPPIII to mitochondria, as measured by colocalization of Trs85-mNeonGreen (Thomas *et al,* 2019) (Figs 5E and EV5). Although a similar Rab11/Ypt31 construct does not recruit TRAPPIII to mitochondria, a chimeric construct containing the NBD of Rab11/Ypt31 and the HVD of Rab1/Ypt1 is able to partially recruit the TRAPPIII complex to mitochondria, suggesting there is a direct physical interaction between the Rab1/Ypt1 HVD and the TRAPPIII complex (Thomas *et al,* 2019) (Figs 5E and EV5). We therefore used ectopic localization of this chimeric construct as an assay to probe the interaction between TRAPPIII and the Ypt1/Rab1 HVD. We created serial alanine patch substitutions of the Rab1/Ypt1 HVD in this construct and tested their ability to recruit TRAPPIII to the mitochondria. We found that substitution of the C-terminal portion, including residues 191–201 of the Rab1/Ypt1 HVD, diminished recruitment of the TRAPPIII complex to the mitochondria (Figs 5F and EV5). These results correlate well with our interpretation of the structure, in which a portion of the HVD corresponding to residues ~180–190 appears disordered, whereas residues ~191–200 appear to be ordered and bound to Trs31.

The Rab1/Ypt1 HVD culminates with two cysteine residues that are prenylated (Pylypenko *et al,* 2006) and insert into the membrane either prior to or concomitant with activation by TRAPPIII. The observed location of the HVD bound to the surface of Trs31 therefore imposes a significant constraint upon any model for how TRAPPIII binds to the membrane surface. This leads us to propose that the complex binds the membrane in the orientation shown in Fig 5A such that the Rab1/Ypt1 NBD is bound to the surface of TRAPPIII that is most distal to the membrane. Additional evidence in favor of

this membrane-bound orientation of the TRAPPIII complex arises from analysis of the electrostatic potential of the surface of the complex, though this analysis has the caveat that some surface residues are unstructured. While there are both positive and negative charges scattered across all surfaces of the TRAPPIII complex, and no single surface contains a concentrated region of positive charge, the proposed membrane-binding surface appears to be the most positively charged of all the surfaces (Fig 5G). In accordance with this idea, we determined that the TRAPPIII complex prefers to bind liposomes with anionic lipid mixtures (Fig 5H and Appendix Table S2). A circumstantial piece of evidence supporting the proposed membrane-bound orientation is the extreme preferred orientation adopted by the complex in vitreous ice (Fig EV2G). These 2D projections (Figs 5B and EV2B) implicate two possibilities for which surface was interacting with the air–water interface within the holes of the EM grid: One is opposite where the Rab1/Ypt1 C terminus is located, while the other is the same surface we propose TRAPPIII uses to bind membranes (Fig 5A). As a peripheral membrane protein, we think it is probable that TRAPPIII interacted with the air–water interface via the same surface it uses to bind to organelle membranes within a cell.

## A conserved amphipathic helix within Trs85 is required for TRAPPIII membrane binding and Rab1 activation

We were concerned that the proposed membrane-binding surface of Trs85 did not appear to possess an overwhelmingly positively charged character, which we expected to be necessary for its interaction with membranes. However, upon closer inspection we identified an unmodeled region on this surface located between the GTPase-like fold and α-solenoid motif (residues 368–409). Though we were unable to model this portion of the protein in our structure because of a lack of clear electron density, the primary sequence is conserved and predicted to fold as an amphipathic α-helix (Fig 6A–C).

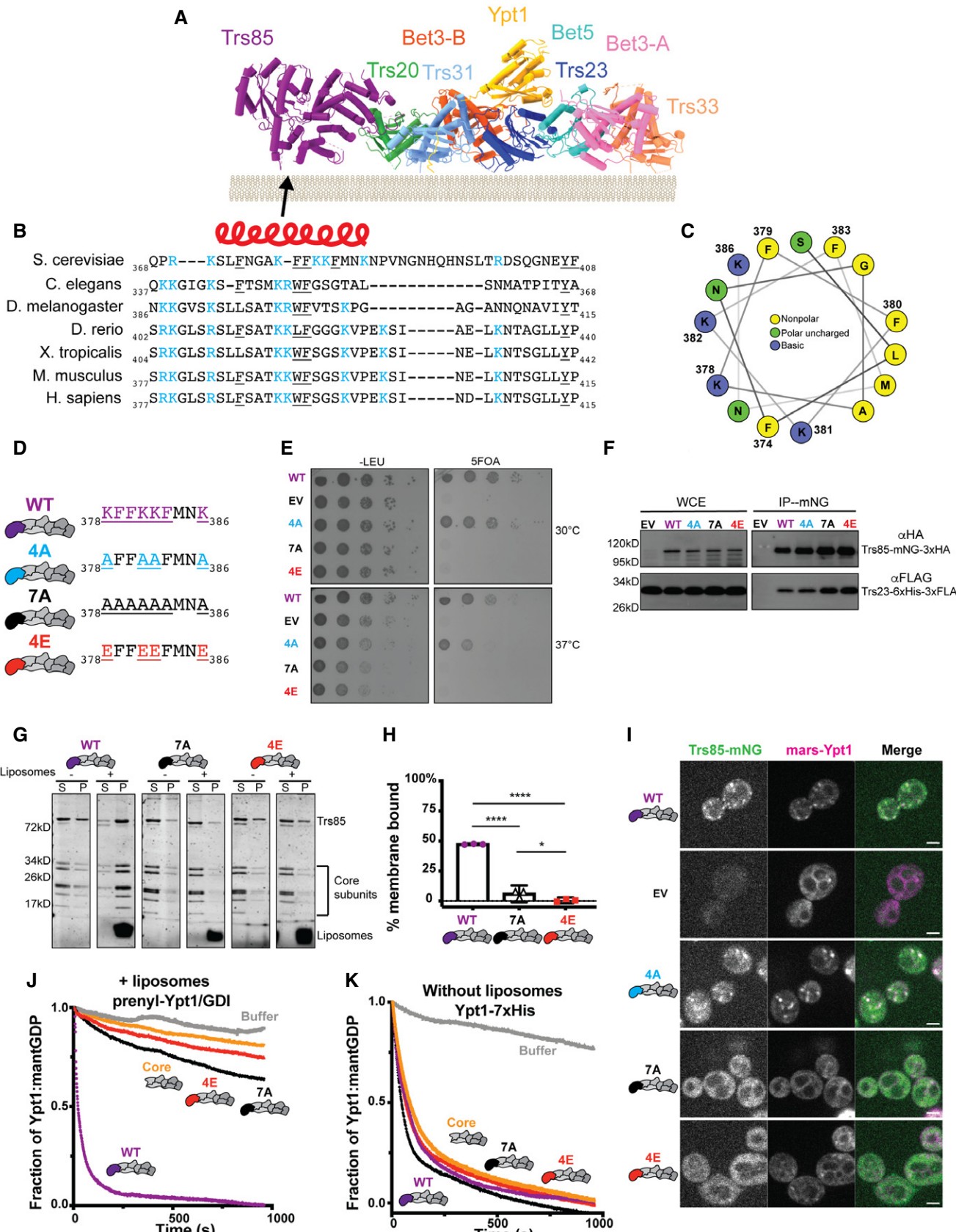

**Figure 6.**

**Figure 6. TRAPPIII uses a conserved amphipathic helix to bind the membrane.**

A Refined atomic model of the TRAPPIII complex shown as it likely interacts with membranes. The arrow points to the location of the unmodeled conserved region.

B Sequence alignment of the unmodeled region predicted to lie on the membrane-binding surface. Red helix above represents the region predicted to fold as an α-helix in secondary structure analysis. Positively charged residues are highlighted in blue, while bulky aromatic residues are underlined.

C Helical wheel representation of the region predicted to be an α-helix. Positively charged residues are colored blue, polar uncharged residues are colored green, and nonpolar residues are colored yellow.

D The wild-type (WT) Trs85 loop sequence is shown on top, with targets of substitution colored purple and specific substitution mutants color coded.

E Yeast complementation assays performed in a sensitized strain (*BET3-GFP*::HIS3, *trs85Δ*::KANMX) to assess the functionality of Trs85 substitution mutants. Representative of $n$ = 3 independent experiments. (WT = wild-type, EV = empty vector).

F Coimmunoprecipitations of Trs23-6xHis-3xFLAG with the same Trs85 substitution mutants. Representative of $n$ = 3 independent experiments (WCE = whole-cell extract, WT = wild-type, EV = empty vector).

G *In vitro* membrane pelleting assay examining two Trs85 substitution mutants in the context of the TRAPPIII complex. In the absence of liposomes, all three complexes pellet slightly; however, in the presence of membranes, only the wild-type Trs85-TRAPPIII complex is strongly enriched in the pelleted fraction. Representative of $n$ = 3 independent experiments (S = supernatant, P = pellet).

H Quantification of (G). Average % protein pelleted between three assays is plotted, and the error bars represent the 95% confidence intervals. The value for each replicate is depicted. The negative values arise from subtraction of the average amount of protein that pellets in the absence of liposomes. Ordinary one-way ANOVA with Tukey's multiple comparisons test: WT versus 7A and WT versus 4E ****$P$ < 0.0001, 7A versus 4E *$P$ = 0.0222.

I Live cell fluorescence microscopy of Trs85-mNeonGreen substitution mutants and mRFPmars-Ypt1. Scale bar = 2 μm. Representative of $n$ = 3 independent experiments. (WT = wild-type, EV = empty vector).

J *In vitro* nucleotide exchange assay of prenylated-Ypt1/GDI substrate in the presence of synthetic liposomes. The wild-type Trs85-TRAPPIII complex (purple) is significantly more active than the substitution mutants (red and black) or the core alone (yellow). Representative of $n$ = 3 independent experiments.

K Assays performed as in (J), but using a non-prenylated Rab1/Ypt1 substrate and in the absence of liposomes. Exchange activity of all complexes is equivalent. Representative of $n$ = 3 independent experiments.

Source data are available online for this figure.

In order to determine whether this putative amphipathic helix was important for Trs85 and TRAPPIII function, we made several substitution mutations in this region, disrupting either the charge or both charge and hydrophobicity (Fig 6D). Substitution of 7 residues in this region with alanine (mutant 7A) resulted in severe loss of function (Fig 6E). Substitution of the basic charge (mutant 4A) displayed slight growth defects, while charge inversion of the lysine residues by substitution with glutamate residues (mutant 4E) resulted in complete loss of function (Fig 6E).

We considered the possibility that these mutations might prevent Trs85 from associating with the core TRAPP subunits, but both the 7A and 4E mutants were stably bound to the TRAPP core in co-IP experiments (Fig 6F). We reasoned that if the 7A and 4E Trs85 substitution mutants associate with the core TRAPP subunits but are not functional, then perhaps these residues are indeed part of an amphipathic helix responsible for mediating TRAPPIII membrane binding, similar to the role of amphipathic helices in many other peripheral membrane proteins (Kahn *et al*, 1992; Antonny *et al*, 1997; Huang *et al*, 2001; Lee *et al*, 2005; Drin & Antonny, 2010). To test this hypothesis, we purified recombinant mutant TRAPPIII complexes and directly tested their relative membrane-binding affinities using the *in vitro* liposome pelleting assay (Fig 6G). Similar to wild-type, a small amount of each complex pelleted in the absence of membranes. However, in striking contrast to wild-type, both the 7A and 4E substitution mutants did not bind to synthetic liposomes (Fig 6H), indicating that this amphipathic helix is required for TRAPPIII membrane binding *in vitro*.

Consistent with a loss of membrane-binding, both the 7A and 4E Trs85 substitution mutants failed to localize to the Golgi and did not activate Rab1/Ypt1 either *in vivo* (Fig 6I) or in biochemical reconstitution assays in which TRAPPIII GEF activity is membrane-dependent (Fig 6J). Importantly, these substitution mutants retained their ability to activate Rab1/Ypt1 in a membrane-independent GEF assay (Fig 6K), indicating that the active site is unaffected by these substitutions. Taken together, these results reveal that the Trs85

amphipathic helix is required for TRAPPIII complex membrane binding and function.

### Similar requirements for TRAPPIII function in secretion and autophagy

Prior work has shown that *trs85Δ* cells display a deficiency in autophagy (Meiling-Wesse *et al*, 2005; Nazarko *et al*, 2005; Lynch-Day *et al*, 2010; Kakuta *et al*, 2012), a defect that can be monitored via the degradation kinetics of GFP-tagged Atg8, the yeast paralog of LC3 (Klionsky *et al*, 2016). In wild-type cells, GFP-Atg8 is degraded as a consequence of rapamycin-induced autophagy, resulting in an accumulation of free GFP. We determined that substitution mutants disrupting the Trs85 amphipathic helix or Trs85 binding to the TRAPP core displayed significant defects in rapamycin-induced GFP-Atg8 degradation (Fig 7A and B). This indicates that the role of the Trs85 amphipathic helix in anchoring the TRAPP core to the membrane surface is critical for both secretion and autophagy (Fig 7C).

## Discussion

GEFs control where and when their GTPase substrates are activated by catalyzing nucleotide exchange. By coupling a localization signal and nucleotide exchange activity within each GEF, the cell can achieve precise spatial and temporal activation of its substrate GTPase at a specific cellular location. The TRAPP complexes activate two of the most important Rab GTPases in the secretory pathway, Rab1 and Rab11. Recent work from our laboratory and others has clarified the roles of TRAPPIII and TRAPPII as GEFs for Rab1 and Rab11, respectively (Jones *et al*, 2000; Meiling-Wesse *et al*, 2005; Morozova *et al*, 2006; Lynch-Day *et al*, 2010; Thomas & Fromme, 2016; Riedel *et al*, 2018; Thomas *et al*, 2018; Thomas *et al*, 2019). Although the activation of Rab1/Ypt1

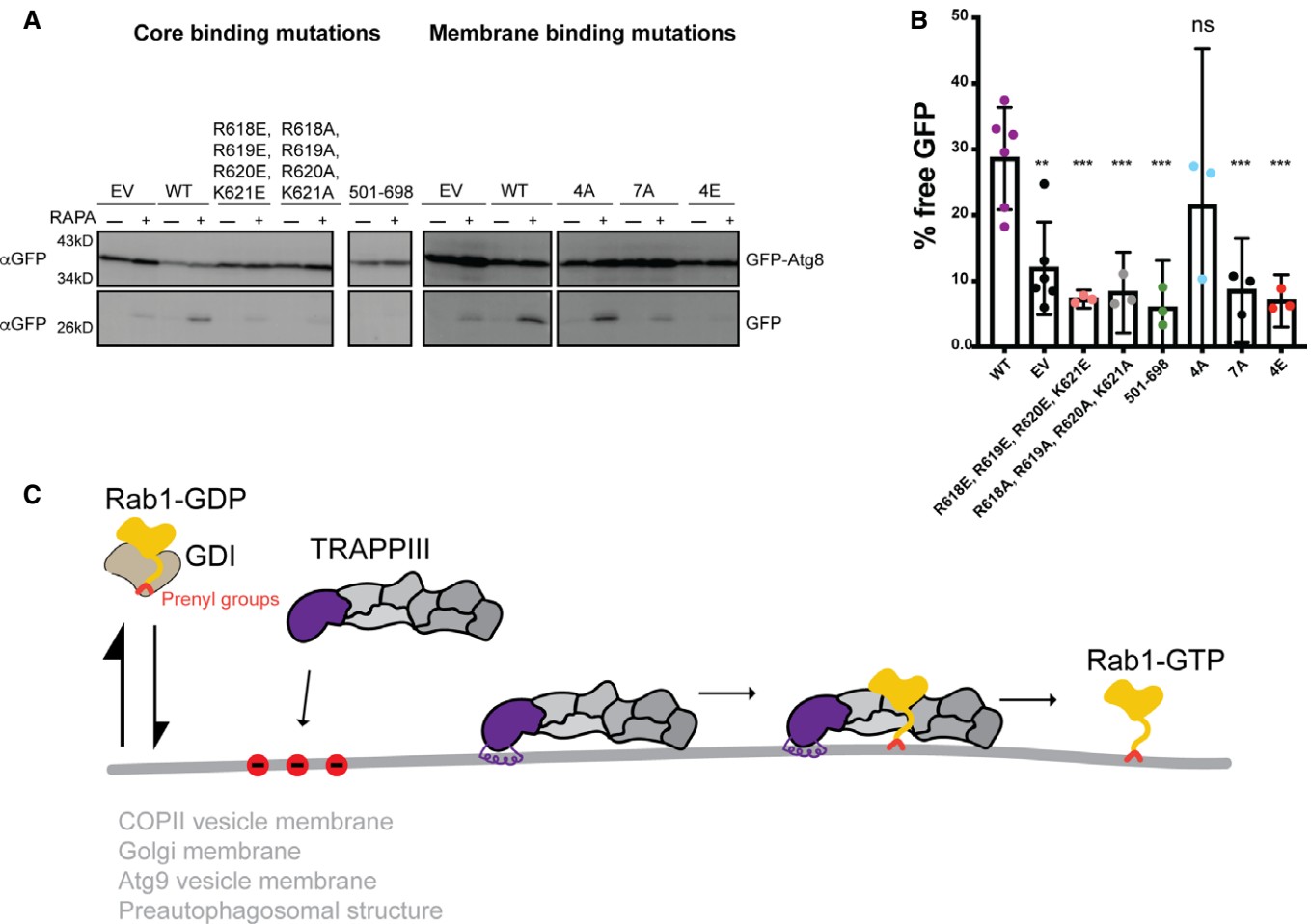

**Figure 7. The membrane-binding helix is critical for autophagy.**

A  GFP-Atg8 processing is monitored for either the TRAPP core binding mutations or the amphipathic helix (membrane binding) substitutions. With functional Trs85, there is an accumulation of free GFP. Samples are from pre-rapamycin treatment (−) and 1hr after addition of rapamycin (+). Representative of $n = 3$ independent experiments. (WT = wild-type, EV = empty vector).

B  Quantification of (A). The average of at least three replicates is plotted with error bars representing the 95% confidence interval. The percent of free GFP for wild-type Trs85 is compared with each mutant using an unpaired, two-tailed Student's *t*-test with Welch's correction: versus EV **$P = 0.0022$, versus R618E, R619E, R620E, K621E ***$P = 0.0008$, versus R618A, R619A, R620A, K621A ***$P = 0.0006$, versus 501–698 ***$P = 0.0003$, versus 4A $P = 0.3295$, versus 7A ***$P = 0.0008$, versus 4E ***$P = 0.0005$, ns = not significant. (WT = wild-type, EV = empty vector).

C  Model for TRAPPIII-mediated Rab1 activation in cells. Inactive Rab1-GDP/GDI exists in an equilibrium where Rab1-GDP samples various organellar membranes. TRAPPIII is recruited to specific membranes containing negative charges (red circles) and utilizes an amphipathic helix to bind stably to the membrane surface. Rab1 is locally activated by TRAPPIII in order to recruit downstream effectors to facilitate trafficking or autophagic events.

Source data are available online for this figure.

by core TRAPP subunits has been extensively studied (Jones *et al*, 2000; Wang *et al*, 2000; Cai *et al*, 2008), the structure and function of the key TRAPPIII regulatory subunit Trs85 have remained elusive. Our results demonstrate that Trs85 serves as a membrane anchor for the TRAPPIII complex by utilizing a conserved amphipathic helix.

The ability of Trs85 to anchor the TRAPPIII complex to a membrane surface is vital to the activation of Rab1/Ypt1 because the stable association of the TRAPPIII complex to the membrane surface establishes proper spatial geometry and increases the frequency of interactions between the active site and the Rab1/Ypt1 substrate (Fig 7C). The observed elevated activity of TRAPPIII

relative to the TRAPP core is due to the fact that the core does not appear able to adequately bind to membranes, and therefore cannot activate Rab1/Ypt1 at the membrane surface.

A key aspect of our work is the presence of the Rab1/Ypt1 HVD in the cryo-EM structure of the TRAPPIII-Rab1/Ypt1 complex. The HVD terminates with two prenylated cysteine residues that serve as membrane anchors for the Rab (Pylypenko *et al*, 2006). At steady state in cells, inactive (GDP-bound) Rabs are bound to GDI proteins that protect the hydrophobic prenyl groups (Goody *et al*, 2005; Wu *et al*, 2010). However, inactive Rabs dynamically dissociate from GDI and transiently bind to membranes before re-associating with GDI. In this way, inactive Rabs are thought to sample different

organelles before encountering their specific GEF to become activated at a particular organelle. The active Rab cannot bind to GDI and is therefore stabilized at its site of activation until inactivated by a Rab-GAP. Accordingly, the structure that we have determined here represents an intermediate of the nucleotide-exchange reaction in which the prenylated cysteines of the HVD are membrane-bound. This structure reveals the HVD is bound by the Trs31 subunit, and its position in the structure provided us with a critical piece of evidence regarding the orientation of the complex on the membrane surface.

Our identification of the TRAPPIII membrane-binding surface was further supported by the presence of a conserved amphipathic helix within Trs85. We found that substitution mutants disrupting either the hydrophobic face of the helix or inverting the charge of the other face completely abrogated membrane binding of the entire TRAPPIII complex, both *in vitro* and *in vivo*.

The fact that we were not able to model the amphipathic helix into the cryo-EM density provides additional, although perhaps counterintuitive, evidence in favor of its role in membrane binding. Amphipathic helices are commonly found on the surfaces of proteins when they do not perform any specialized function but rather are simply part of the protein fold in which hydrophobic side-chains are on the interior and hydrophilic sidechains are on the exterior of the folded protein. These α-helices should be readily visible in the structure, as are all of the other predicted α-helices in the TRAPPIII complex structure. However, an amphipathic α-helix that has evolved to insert into the membrane would not be expected to form part of the hydrophilic surface of a folded protein domain (Drin & Antonny, 2010). Rather, it should extend from the globular portion of the protein in order to insert longitudinally into the outer leaflet of the membrane, as is the case with the Trs85 amphipathic α-helix that is required for TRAPPIII membrane binding.

Although there are many examples of amphipathic helices serving as membrane anchors for smaller proteins, such as Arf GTPases and N-BAR domain-containing proteins (Drin & Antonny, 2010), TRAPPIII represents a notable example in which a sizeable multiprotein complex is anchored by a single amphipathic helix. Remarkably, the substitution of just four residues in this helix was sufficient to disrupt the binding, and therefore function, of the entire 250 kD complex.

The core subunits likely provide additional protein–protein interactions or perhaps supplementary interactions with organelle membranes, as Trs85 localization to the Golgi was abrogated in cells when substitutions that diminished its binding to the core were introduced. We determined that the TRAPPIII complex prefers to bind to membranes with a significant anionic charge, but did not identify a specific lipid requirement (Fig 5F and Appendix Table S2). Given that TRAPPIII has been implicated to function at many locations within the cell including on COPII vesicles, at the Golgi, on Atg9 vesicles, and at the preautophagosomal structure, it is likely that additional protein binding partners help specify TRAPPIII localization *in vivo* via coincidence detection. For example, TRAPPIII localization to COPII vesicles likely depends upon an interaction between the TRAPP core and the COPII coat subunit Sec23 (Cai *et al*, 2007; Lord *et al*, 2011). Future studies of such functions of TRAPP core subunits are important but made challenging by the need to disentangle observed roles of these subunits in TRAPPIII versus TRAPPII.

Many questions remain regarding the role of both TRAPPIII and TRAPPII in autophagy. Our results indicate that the key function of Trs85, serving as a membrane anchor for the TRAPPIII complex, is important for both normal growth and autophagy.

# Materials and Methods

Plasmids and yeast strains used are presented in Appendix Tables S3 and S4.

### TRAPPIII purification

The full set of TRAPPIII complex genes from *Saccharomyces cerevisiae* were cloned into two plasmids. Two copies of TRS33 were included to compensate for the otherwise substoichiometric expression levels of this subunit. *TRS85* with an N-terminal hexahistidine tag and *TRS33* were inserted into a pETDuet-1 vector (CFB2868). As reported previously, *BET3*, *BET5*, *TRS20*, *TRS23*, *TRS31*, and *TRS33* were inserted into a pCOLADuet-1 vector (pLT21, (Thomas & Fromme, 2016)). These plasmids were co-transformed into the Rosetta2 strain of *E.coli* (Novagen). Multiple liters of culture were grown in terrific broth for 8–12 h at 37°C (until the $OD_{600}$ ~2–3). The temperature was then reduced to 16°C and 1 h later IPTG was added to a final concentration of 300 μM to induce protein expression. Following overnight expression (14–18 h), cells were collected via centrifugation and resuspended in lysis buffer (40 mM Tris pH 8, 300 mM NaCl, 10% glycerol, 10 mM imidazole, 1 mM DTT, 1 mM PMSF, and 1× Roche PIC). Cells were lysed by sonication, and after clearing the lysate, the complex was purified by nickel affinity chromatography (Ni-NTA resin, Qiagen) and eluted with elution buffer (40 mM Tris pH 8, 300 mM NaCl, 10% glycerol, 250 mM imidazole, 1 mM DTT). The sample was further purified using anion exchange chromatography with a linear gradient on a MonoQ column (GE Healthcare) (Buffer A: 20 mM Tris pH 8, 1 mM DTT; Buffer B: 20 mM Tris pH 8, 1 M NaCl, 1 mM DTT). The peak fractions were collected, concentrated, and frozen in liquid nitrogen prior to storage at −80°C. The fractions were thawed (or never frozen) and concentrated for size exclusion chromatography on a Superdex200 10/300 (GE Healthcare) preequilibrated in 10 mM Tris pH 8, 150 mM NaCl, 1 mM DTT.

### Ypt1 purification

GST-Ypt1-7xHis (gift from T. Bretscher laboratory) or GST-Ypt1 (pLT50, (Thomas & Fromme, 2016)), with a cleavable N-terminal GST tag in the pGEX-6P vector backbone, was transformed into the Rosetta2 strain of *E. coli* (Novagen). 1–4 l of culture were grown in terrific broth for 8–12 h at 37°C (until the $OD_{600}$ ~2–3). The temperature was then reduced to 16°C and 1 h later IPTG was added to a final concentration of 300 μM to induce protein expression. Following overnight expression (14–18 h), cells were collected via centrifugation and resuspended in lysis buffer (1× PBS, 5 mM BME, 2 mM $MgCl_2$). Cells were lysed by sonication, and the clarified lysate was incubated with Glutathione resin (G-Biosciences) for 2–3 h at 4°C to isolate GST-tagged proteins. The resin was washed in lysis buffer, resuspended in PreScission cleavage buffer (50 mM Tris pH 7.5, 150 mM NaCl, 1 mM EDTA, 2 mM $MgCl_2$, and 1 mM DTT), and

treated overnight with PreScission (3C) protease (~40 µl at 1.3 mg/ml) at 4°C to remove the GST tag. Cleavage of the GST tag eluted the proteins from the resin. After treatment, the supernatant was collected and analyzed by SDS–PAGE.

### TRAPPIII-Rab1/Ypt1 complex formation

Purified TRAPPIII complex (from anionic exchange step) and Ypt1-7xHis were mixed in a 1:3 molar ratio and incubated overnight at 4°C with calf intestinal alkaline phosphatase (Sigma) to hydrolyze nucleotide. After 12–18 h, the sample mixture was subjected to size exclusion chromatography on a Superdex 200 Increase 10/300 column (GE Healthcare) in EM buffer (10 mM Tris pH 7.5, 350 mM NaCl, 1 mM DTT). The trace displayed two dominant peaks corresponding to TRAPPIII-Rab1/Ypt1 complex and free Rab1/Ypt1 (Fig EV1A). The fractions containing the complex were pooled and concentrated in a 100 kDa molecular weight cutoff concentrator (Millipore). The protein was then used immediately to prepare grids for cryo-EM.

### Cryo-EM sample preparation, data collection, and data processing

The complex, at 0.1–0.3 mg/ml in EM buffer containing either 0.02% Tween-20 or 0.025% amphipol A8-35 (Anatrace), was applied to Quantifoil or UltrAufoil R1.2/1.3 grids (Quantifoil) then blotted for 2 s at 4°C with 100% humidity and plunged into liquid nitrogen-cooled liquid ethane using a Vitrobot IV (TFS).

#### "Flat" data

Initial imaging data were collected at the MRC-LMB on a Titan Krios operated at 300kV using a Falcon-III detector and 59kX nominal magnification resulting in a 1.34 Å/pixel size on the detector. A total of 809 fractionated exposures were obtained over two different 24-h data collection sessions. RELION versions 2, 3, and 3.1 (Scheres, 2012; Kimanius et al, 2016; Zivanov et al, 2018; Zivanov et al, 2020) were used for all data processing steps. MotionCor2 (Zheng et al, 2017, p. 2) was used to perform motion-correction and dose weighting, and GCTF (Zhang, 2016) was used to estimate micrograph defocus values. After triaging micrographs, particles were autopicked and 2D classification was used to remove junk particles. The negative-stain structure of the TRAPPIII complex (Tan et al, 2013) was initially used as a reference model for 3D refinement. After iterative rounds of 3D classification and 3D refinement, it was evident that the resulting 3D reconstructions were quite anisotropic due to a strong orientation bias of the particles. Alternative particle-picking strategies did not provide additional orientations, and therefore, we decided to collect data from tilted grids.

#### 30°-tilt data

A dataset of 430 micrographs was collected on the same microscope with the stage tilted to 30°. An additional 2015 micrographs were collected at the Diamond Light Source / eBIC on a Titan Krios operated at 300 kV with a K2 detector and energy filter (Gatan). The image pixel size on the K2 detector was 0.824 Å/pixel after binning super-resolution images. Data were processed as described above for the "flat" data, with an additional step of per-particle defocus estimation using GCTF prior to 2D classification.

Each dataset was initially processed independently, producing reconstructions of ~4.5 Å resolution using the 0.143 FSC criterion. The datasets were then processed together, subjected to multiple 3D classification and 3D refinement procedures, and after CTF-Refinement and Bayesian Polishing ultimately resulted in a 3D reconstruction with a 0.143 FSC resolution of 3.8 Å. However, this 3D reconstruction was still fairly anisotropic and interpretation of the distal ends of the map was further confounded by some flexibility of the complex that could not be surmounted by focused local refinements or classifications. We therefore decided to collect additional data at even higher tilt angles, as the Lyumkis group demonstrated that 40° tilt was feasible for data collection (Tan et al, 2017).

#### 40°-tilt and 45°-tilt data

The Cornell CCMR facility Talos Arctica (TFS) with K3 detector and BioQuantum energy filter (Gatan) was used to collect 3,062 micrographs on 45°-tilted grids and 1950 micrographs on 40°-tilted grids. The microscope was operated at 200 kV at a nominal magnification of 63kX resulting in an image pixel size of 1.24 Å/pixel. Only a small fraction of these micrographs was suitable for use, either due to extensive beam-induced motion that could not be corrected, or insufficient contrast. cryoSPARC (Punjani et al, 2017) was used to triage micrographs after "Patch-CTF" micrograph estimation. cryoSPARC was also used for 2D classification and initial 3D classifications to generate a clean particle set that was then imported into RELION with the use of the csparc2star python script written by D. Asarnow (https://github.com/asarnow/pyem). RELION 3.1 was used to combine the particles imaged at 30°, 40°, and 45°, using the scheme shown in Fig EV3. Extensive 3D classification was used to select the best set of particles. In order to deal with apparent flexibility of Trs85 relative to the rest of the complex, we employed Trs85-focused particle subtraction and 3D classification. CTF refinement and Bayesian polishing were also used, culminating in a final 3D reconstruction from 69,315 particles with a 0.143 FSC resolution of 3.7 Å and 0.5 FSC resolution of 4.2 Å.

Density modification using RESOLVE (Terwilliger et al, 2020) in Phenix (Liebschner et al, 2019) was used to facilitate interpretation of the reconstruction. The resulting map was reported by RESOLVE to have a resolution of 3.5 Å. In addition, focused local refinement was performed separately on Trs85, the distal region surrounding Trs33, and the central region surrounding Rab1/Ypt1 to generate additional maps to assist with model building.

Particle orientation distributions were assessed and plotted using scripts from A. Leschziner (https://github.com/leschzinerlab/Relion) and D. Lyumkis et al (https://github.com/nysbc/Anisotropy).

### Model building and refinement

Available crystal structures (Kim et al, 2005, 2006; Cai et al, 2008) for the core subunits and Rab1/Ypt1 were docked into the map and rebuilt using Coot (Emsley et al, 2010). trRosetta (Yang et al, 2020) was used to generate models of Trs85, and portions of these models were docked or used as guides for de novo model building. Phenix was used for real-space refinement (Afonine et al, 2018; Liebschner et al, 2019). Restrained molecular dynamics simulations were performed using ISOLDE (Croll, 2018) within ChimeraX (Goddard et al, 2018) in order to identify and fix model errors arising due to imposition of Ramachandran restraints during refinement. The final

refinement was carried out using one half-map, and validation was performed using the other half-map. The model-map 0.143 FSC values were 3.2 Å (whole-map), 3.2 Å (refinement half-map), and 3.4 Å (validation half-map). Model-map FSC curves are shown in Fig EV2F, and model validation statistics (Afonine *et al*, 2018) are shown in Table 1. PyMol (Schrodinger), Chimera (Pettersen *et al*, 2004), and ChimeraX were used to produce images for figures.

### Surface analyses

Conservation analysis was carried out using the Consurf web server utilizing custom alignments generated with Clustal Omega (Armon *et al*, 2001; Landau *et al*, 2005; Ashkenazy *et al*, 2010; Celniker *et al*, 2013; Ashkenazy *et al*, 2016). The electrostatic surface potential was calculated using the APBS plugin for Pymol (Schrodinger), and the map was colored on a red (−5kT/e) to blue (+5kT/e) scale (Baker *et al*, 2001; Dolinsky *et al*, 2007).

### Sequence alignment

The multiple sequence alignments were performed using Clustal Omega with default settings (Madeira *et al*, 2019) and then manually adjusted for fine-tuning of the amphipathic helix region.

### Cross-linking mass spectrometry (XL-MS)

~100 µg of purified TRAPPIII-Rab1/Ypt1 complex (in 10 mM HEPES pH 8, 150 mM NaCl, 1mM DTT) was incubated on ice for 1 h with a range of disuccinimidyl sulfoxide (DSSO) concentrations spanning 0.3125 mM to 2.5 mM. The reaction was then quenched with 10mM Tris pH 8 for 15 min at room temperature. 10% of the sample was analyzed by SDS–PAGE (Fig EV1B), and the sample cross-linked with 625 µM DSSO was chosen for further analysis. To prepare the sample for mass spectrometry, the cross-linked proteins were first dried completely in a vacuum concentrator and then resuspended in 8 M Urea, 50 mM Tris pH 8, and 5 mM DTT. Urea-solubilized proteins were then incubated at 37°C for 30 min for denaturation and to reduce disulfide bridges. Reduced cysteines were blocked with 25 mM iodoacetamide for 15 min. Urea was then diluted to 2 M with a solution of 50 mM Tris pH 8 and 150 mM NaCl. Proteins were digested overnight with gentle nutation at 37°C using 1 µg of Promega Trypsin GOLD®. The next day, the digested sample was acidified with 0.25% formic acid + 0.25% trifluoroacetic acid and desalted using a Waters Sep-Pak column. The desalted, digested sample was dried in a vacuum concentrator and reconstituted in 0.1% TFA + 1 picomole/µl Angiotensin-II peptide (Sigma). The sample was subjected to LC-MS/MS/MS analysis on an Orbitrap Fusion Lumos mass spectrometer (Thermo-Fisher Scientific). On-line sample fractionation was performed with linear gradients of 0.1% formic acid (solvent A) and 80% acetonitrile + 0.1% formic acid (solvent B) at a flow rate of 300 nl/min using a home-made 125-µm × 25-cm capillary column packed with 3-µm C18 resin. Precursor ions were detected in the Orbitrap mass analyzer (375–1,500Th, resolution 60,000 at 375Th). Charge states in the 4+ to 8+ range were selected for MS2 analysis in the Orbitrap (resolution set at 30,000 at 375Th) with 25% CID energy. Peaks with a mass difference of 31.9721 Da, indicative of a DSSO cross-link, were subjected to MS3 analysis. For MS3, ions were

fragmented in the linear ion trap using higher-energy collisional dissociation (HCD) at 35% HCD energy. The cross-link search was performed using MaXLinker software (Yugandhar *et al*, 2020). The structure mapping of cross-links was performed using Xlink Analyzer (Kosinski *et al*, 2015) implemented in UCSF Chimera (Pettersen *et al*, 2004). Figs 2C, 4D, and 5C were generated using xVis (Grimm *et al*, 2015).

### Liposome preparation

Synthetic liposomes were prepared using a mixture of lipids (Appendix Table S2). Lipids were combined, dried by rotary evaporation, rehydrated overnight in HK buffer (20 mM HEPES pH 7.4, and 160 mM KOAc) at 37°C, then extruded through 100 or 400 nm filters (Whatman) to generate homogenous liposomes. Synthetic liposomes were stored at 4°C until needed.

### Pelleting assays

In 40 µl reactions, loaded ~1–2 µg of protein (300 nM) along with 500 µM of 400 nm liposomes in HK buffer. Incubated for 10 min at RT. Samples were ultracentrifuged using a TLA100.3 rotor (Beckman Coulter) at roughly 130,000 *g* for 15 min at 4°C. The supernatant and pellet fractions were recovered and evaluated separately via SDS–PAGE. Each assay was performed at least three times. The gels shown are representative of all three experiments. The percent of membrane-bound protein is calculated by dividing the intensity of the pelleted bands by the total intensity of protein in both the supernatant and pelleted fractions. Then, the average amount of protein that pellets in the absence of liposomes is subtracted from each sample. The quantification is based on the averages of all three experiments. All direct comparisons were performed using the same batch of liposomes to account for batch variability. Gel densitometry was performed using FIJI (Schindelin *et al*, 2012).

### 5-FOA complementation assay

Corresponding plasmids harboring mutant versions of Trs85 were transformed into a sensitized strain background (*trs85Δ*::KAN, Bet3-GFP::HIS (CFY3307)), grown on -LEU plates. Several colonies were resuspended in water then serially diluted and replicated onto -LEU and 5FOA plates. Cells were grown for 48 h at 26, 30, or 37°C.

### Microscopy

For localization experiments, plasmids harboring mutant versions of Trs85 were transformed into a *trs85Δ*::KANMX strain (CFY2692) so that only the plasmid copy of Trs85 would be present. For experiments assessing Rab1/Ypt1 localization, a second plasmid containing mRFPmars-Ypt1 (CFB3511) was co-transformed. For mitochondrial anchor-away assays, plasmids harboring nucleotide-free Rab-Fis1 fusions were transformed into a strain with endogenously tagged Trs85-mNeonGreen (CFY2449).

Cells were grown at 30°C until log-phase. Cells were collected by centrifugation for 2 min at 12,000 *g*, resuspended in a small amount of media, and placed on a microscope slide with coverslip. Cells were imaged at room temperature with a CSU-X spinning-disk

confocal microscope system (Intelligent Imaging Innovations) on a DMI6000 microscope (Leica Microsystems) equipped with a CSU-X1 spinning-disk confocal unit (Yokogawa Electric Corporation) and a QuantEM 512SC camera (Photometrics). The objective was a $100 \times 1.46$ NA Plan Apochromat oil immersion lens (Leica Microsystems). Additional components included a laser stack and mSAC (spherical aberration correction; Intelligent Imaging Innovations). SlideBook software was used to control the system (Intelligent Imaging Innovations) and for data analysis.

All images within a figure panel had their minimum or maximum brightness levels adjusted equivalently for clarity. The images shown are representative of all experiments. The Mander's Overlap Coefficient (MOC) was calculated, using the JACOP plugin (Bolte & Cordelières, 2006) in FIJI (Schindelin et al, 2012), by analyzing cropped image stacks containing between 2 and 9 cells. These cropped images were thresholded so that the analysis focused on mitochondrial and punctate signals but avoided the cytoplasmic background haze. The average MOC was calculated based on the MOC from at least 17 unique images and encompassed a minimum of 83 cells. Statistical differences were evaluated using a one-way ANOVA with a Tukey's multiple comparisons test in Prism (GraphPad).

## Co-immunoprecipitations

Plasmids harboring Trs85 mutants were transformed into a yeast strain containing trs85Δ::KANMX, Trs23-6xHis-3xFLAG::TRP (CFY3585). Overnight cultures were grown to saturation and then used to inoculate fresh media. Large cultures grew until log-phase and then cells were collected by centrifugation. Cell pellets were resuspended in 1 ml cold water and then transferred to an Eppendorf tube. Cell pellets were aliquoted to approximately equal weights, and then, the cells were collected again, flash-frozen in liquid nitrogen, and stored at $-80°C$. Each coIP contained roughly 25 $OD_{600}$ of cells per tube.

Upon removal from storage, 250 μl of lysis buffer [50mM Tris pH 7.5, 0.2% NP40, 150mM NaCl, 1mM EDTA] and 100 μl of 0.5 mm glass beads were added to each tube. Cells were immediately lysed via harsh vortexing for $2 \times 10$ min at 4°C, with 2 min rest on ice in between. Lysed cell debris was pelleted for 5 min at $21,000 \times g$ at 4°C, and the supernatant was collected. 25 μl supernatant was used for SDS–PAGE analysis as the whole-cell extract. The remaining lysate was incubated with 12.5 μl equilibrated magnetic mNeon-Green-Trap resin (Chromotek) for a minimum of 1 h rotating at 4°C. After incubation, the resin was collected and washed $3 \times 1$ ml with lysis buffer and transferred once to a new tube. Finally, 50 μl of 2× SDS sample buffer was added to the resin and incubated at least 10 min at 95°C. Samples were subjected to SDS–PAGE and transferred to a PVDF membrane (Millipore) for Western blot analysis probing for HA and FLAG epitopes using 1° antibody = 1:500 mouse αHA (Roche), or 1:500 mouse αFLAG (Roche) and 2° 1:5,000 sheep-αmouse (Amersham).

## GFP-Atg8 processing assay

Plasmids harboring Trs85 mutants were co-transformed along with GFP-Atg8 (CFB2250) into trs85Δ::KAN (CFY2692). Overnight cultures were grown to saturation at 30°C and then used to inoculate fresh media. Once back in log-phase, ~5–10 $OD_{600}$ units of cells

were collected by centrifugation, 3 min at ~3k rpm and processed for TCA precipitation. With the remaining culture, autophagy was induced by the addition of rapamycin to a concentration of 0.2 μM. After 1 h of incubation at growth temperature, an additional ~5–10 $OD_{600}$ units of cells were collected and processed for TCA precipitation. Samples were resolved by SDS–PAGE and then subjected to Western blot analysis using 1°, 1:1,000 mouse αGFP (Roche) and 2°, 1:5,000 sheep αmouse (Amersham). Autophagy efficiency is quantified measuring the amount of free GFP compared with the total amount of GFP-Atg8 and GFP. The blots shown are representative of all three experiments. The percent of free GFP is calculated by dividing the intensity of the free GFP band by the total intensity of both GFP-Atg8 and free GFP in the rapamycin-induced lane. The quantification is based on the averages of all three experiments. Gel densitometry was performed using FIJI (Schindelin et al, 2012).

## Prenylated Rab1/Ypt1 preparation

Preparation of prenylated-Ypt1/GDI complex was performed as described previously (Thomas et al, 2018). Briefly, Rab/GDI substrates, the mantGDP-loaded Rab, Gdi1, $His_6$-Mrs6, and $His_6$-Bet2/Bet4 were mixed in a 10:10:1:1 molar ratio with a sixfold excess of geranylgeranyl pyrophosphate (Cayman) in prenylation buffer with 20 μM mantGDP and incubated at 37°C for 1 h. Then, imidazole was added for a final concentration of 10 mM. The solution was incubated with a small volume of Ni-NTA resin (Qiagen) at 4°C for 1 h to remove enzymes. Stoichiometric Rab/GDI complexes were then isolated using size exclusion chromatography.

## Rab1/Ypt1 nucleotide loading for GEF assays

~1 mg of purified Rab1/Ypt1 (either prenylated-Ypt1/GDI complex or non-prenylated Ypt1-7xHis) was incubated in the presence of 20 μM mantGDP (Sigma) and EDTA to facilitate exchange for 30 min at 30°C. Afterward, protein was buffer exchanged in 15× excess buffer to remove leftover nucleotide and EDTA before being aliquoted and stored at $-80°C$.

## *In vitro* nucleotide exchange (GEF) assays

For all assays, nucleotide exchange was measured by quenched fluorescence of mantGDP (Sigma) (360 nm excitation and 440 nm emission). Reactions were performed inside of a fluorometer (Photon Technology International) at 30°C with the sequential addition of: 333 μM synthetic liposomes (100 nm), 200 μM nonfluorescent GTP, and ~250 nM prenylated mantGDP:Ypt1 to HKM buffer (20 mM HEPES pH 7.4, 160 mM KoAC, 2 mM $MgCl_2$). The mixture was then incubated at 30°C for 2 min to allow the Rab/GDI substrate time to equilibrate with the membrane. Nucleotide exchange was monitored after the addition of 20 nM GEF complex. For soluble mantGDP:Ypt1-7His, identical conditions were utilized except the liposomes were replaced by HKM buffer.

## Software

Structural biology software is maintained in our laboratory by SBGrid (Morin et al, 2013).

# Data availability

Structural data have been deposited in the RCSB PDB (https://www.rcsb.org) and EMDB (https://www.ebi.ac.uk/pdbe/emdb) databases; the accession codes are 7KMT and EMD-22928, respectively. Mass-spectrometry data has been deposited in the Proteomics Identification Database (PRIDE, https://www.ebi.ac.uk/pride/) as entry PXD025172.

**Expanded View** for this article is available online.

## Acknowledgements

We acknowledge the MRC-LMB Electron Microscopy Facility, Diamond Light Source, and Cornell Center for Materials Research (CCMR) for access and support of electron microscopy sample preparation and data collection. We particularly thank Giuseppe Cannone, Rebecca Voorhees, Mariena Silvestry-Ramos, and Katherine Spoth for assistance with cryo-EM data collection. We also thank Bridget Carragher and Zhening Zhang of the NYSBC, and Chen Xu of the UMass Cryo-EM facility, for performing preliminary cryo-EM data collection. We thank Uche Chukwukere, Jeffrey Ho, and Kelly Rosch for aiding with preliminary cell biological experiments, Ting-Yi Wang for assisting with mass spectrometer maintenance, and Laura Thomas for generating preliminary biochemical data. We thank Daniel Asarnow, Andres Leschziner, and Dmitry Lyumkis for making scripts publicly available on GitHub. This work was supported by National Institutes of Health grants R01GM097272 and R01HD095296 to M.B.S., R01GM116942 and R35GM136258 to J.C.F., R01GM124559, and R01GM125639 to H.Y., by an Alfred P. Sloan Foundation Fellowship and a National Science Foundation Graduate Research Fellowship (grant DGE-1650441) to A.M.N.J., by NSF grant DBI-1661380 to H.Y., and by UK Medical Research Council (MRC_UP_1201/10) funding to E.A.M. Any opinions, findings, conclusions, or recommendations expressed in this material are those of the authors and do not necessarily reflect the views of the funders. This work made use of the Cornell Center for Materials Research Shared Facilities which are supported through the NSF MRSEC program (DMR-1719875). We acknowledge Diamond for access and support of the Cryo-EM facilities at the UK national electron bio-imaging center (eBIC), proposal EM17434, funded by the Wellcome Trust, MRC, and BBSRC.

## Author contributions

Conceptualization and design, data acquisition/investigation (cryo-EM, XL-MS, fluorescence microscopy, cell-based assays, *in vitro* assays), data analysis/interpretation, and writing: AMNJ; Data acquisition (cryo-EM): BPP; Data acquisition (XL-MS): KY; Data acquisition (XL-MS): EJS; Funding acquisition and data interpretation: MBS; Funding acquisition and data interpretation: HY; Funding acquisition and data interpretation: EAM; Funding acquisition, conceptualization and design, data analysis/interpretation, and writing: JCF.

## Conflict of interest

The authors declare that they have no conflict of interest.

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
