## [Review Process File · The EMBO Journal]

Structural basis of TRAPPIII-mediated Rab1 activation

Aaron Joiner, Ben Phillips, Kumar Yugandhar, Ethan Sanford, Marcus Smolka, Haiyuan Yu, Elizabeth Miller, and J. Christopher Fromme

DOI: [10.15252/embj.2020107607](https://doi.org/10.15252/embj.2020107607)

Corresponding author(s): J. Christopher Fromme (jcf14@cornell.edu)

Review Timeline:

Submission Date:	29th Dec 20
Editorial Decision:	9th Feb 21
Revision Received:	11th Mar 21
Editorial Decision:	23rd Mar 21
Revision Received:	7th Apr 21
Accepted:	11th Apr 21

Editor: Elisabetta Argenzio

Transaction Report:

Thank you for submitting your manuscript entitled "Structure and mechanism of TRAPPIII-mediated Rab1 activation" (EMBOJ-2020- 107607) to The EMBO Journal. Your study has now been assessed by three reviewers, whose reports are enclosed below for your information.

As you can see, the referees find your work interesting, but also suggest that you address a few points in order to strengthen the main conclusions.

Given the overall interest of your study, we have decided to invite you to submit a new version of the manuscript revised according to the referees' requests. I should add that it is The EMBO Journal policy to allow only a single round of revision, and acceptance of your manuscript will therefore depend on the completeness of your responses in the revised version.

We generally grant three months as standard revision time. As we are aware that many laboratories cannot function at full capacity owing to the COVID-19 pandemic, we may relax this deadline. Also, we have decided to apply our 'scooping protection policy' to the time span required for you to fully revise your manuscript and address the experimental issues highlighted herein. Nevertheless, please inform us as soon as a paper with related content is published elsewhere.

When preparing your letter of response to the referees' comments, please bear in mind that this will form part of the Review Process File and will therefore be made available online. For more details on our Transparent Editorial Process, please visit our website:

http://emboj.embopress.org/about#Transparent_Process

Before submitting your revised manuscript, deposit any primary datasets and computer code produced in this study in an appropriate public database (see <http://msb.embopress.org/authorguide#dataavailability>). Please remember to provide a reviewer password, in case such datasets are not yet public. The accession numbers and database names should be listed in a formal "Data Availability" section (placed after Materials & Method). Provide a "Data availability" section even if there are no primary datasets produced in the study.

Feel free to contact me if you have any questions about the submission of the revised manuscript to The EMBO Journal. I thank you again for the opportunity to consider this work for publication and look forward to your revision.

Referee #1:

The authors present a well-written, comprehensive, and multidisciplinary analysis of the TRAPPIII complex (a GEF) and its interactions with Rab1/Ypt1 (a key regulatory GTPase in the secretory and autophagic pathways) and with membranes. While the linchpin of the story is the cryoEM structure of a TRAPPIII-Ypt1 complex (a noteworthy achievement in its own right), the impact of this structure is enhanced by an expansive array of complementary biochemical and in vivo experiments. Together, these reveal the binding site for the key juxtamembrane 'hypervariable'

region of the Rab which, in turn, suggests the orientation of the entire TRAPPIII complex on membranes. This hypothesis leads the authors to propose, and then experimentally confirm, that a disordered region of the TRAPPIII-specific subunit Trs85 forms a membrane-inserting amphipathic alpha-helix. The Trs85 amphipathic helix, by helping anchor the TRAPPIII complex to membranes in the proposed orientation, plays an essential role in Rab1 activation.

Taken together, these findings represent a major advance in our understanding of Rab1 biology. The story is mechanistically interesting and biologically important. Overall, while there are always further experiments one could propose, I think the manuscript in its current form is suitable in completeness and impact to merit publication without substantial revision.

Referee #2:

Using a cryo-EM based approach the authors report the structure of the yeast TRAPPIII complex bound to its cognate GTPase Ypt1. TRAPPIII (is one of two yeast TRAPP complexes - the other being TRAPPII) functions on at least three distinct membrane populations (COPII vesicles, the Golgi, and on autophagosomal membranes). The TRAPPs function as Ypt / Rab GEFs and share the same "core" components whilst containing compositionally distinct subunits that presumably mediate Ypt/Rab-specific interactions as well as membrane associations. In the case of TRAPPIII the distinct component is termed Trs85p (in yeast).

This is detailed and compressive study in which not only novel structural insight is provided, but also importantly, functional insight in cells. Through their structural studies the authors identify how Ypt1 associates with TRAPPIII and in so doing define the orientation in which TRAPPIII binds to membranes. In addition, the authors also establish that an evolutionarily conserved amphipathic helix in Trs85 is instrumental in mediated TRAPPIII (and Trs85) association with anionic lipid containing liposomes and membranes in cells

I am not an expert in structural studies but nevertheless I would expect this study to be of significant interest to cell biologists in general, and to those who studying membrane trafficking mechanisms in particular (including autophagy).

The manuscript is well written and laid out in a manner which guides the reader.

Minor points to be addressed:

Figure 4 Panel B - the protein(s) migrating between 75 and 95kDa are labeled as non-specific but are of a similar MW to Trs85 and the Trs85 variant shown in panels A and C (respectively). So, presumably these non-specific proteins are also present in panels A and C. This should be mentioned in the main text (or expanded upon in the corresponding figure legend) to avoid confusion when future readers are scrutinizing the figure.

Page 9. Would the authors please describe more completely (in genetic terms) what they mean by "sensitized background" with reference to Bet3.

Although it is becoming increasing common in the literature to see references to amino acid substitutions as mutated amino acids - this does not (in this reviewer's opinion) make sense - but is

rather lab parlance / jargon. The authors should be encouraged to make changes to the main text to rectify this. For example, on the top of page 14.

Referee #3:

This paper takes a cryo-EM structural approach to examine yeast TRAPP^{III}, the GEF that activates the Rab1 homolog Ypt1 to regulate the early secretory pathway as well as autophagy. The structure reveals the binding site for the Rab1/Ypt1 hypervariable domain. Trs85 is the subunit that defines the yeast TRAPP^{III} version of the TRAPP complex, and the authors report that Trs85 apparently has a conserved amphipathic helix that anchors TRAPP^{III} in the membrane. This work extends previous structural and functional analysis of the core TRAPP complex, and it sheds light on mutations that compromise the function of TRAPP^{III}.

At 3.7 Å, the resolution of this structure is modest. The authors explain how the orientation bias of the complex precluded a higher resolution analysis, and how they used tilted grids to partially overcome this limitation. The core subunit structures are known from crystallography, while the Trs85 subunit structure was predicted with fairly high confidence and shows a reasonable fit to the cryo-EM data. Cross-linking mass spectrometry was used to validate the structure, so overall the results seem to be solid.

The authors push the system to argue for a specific interaction of the Rab1/Ypt1 hypervariable domain with the TRAPP^{III} complex, and for the presence in Trs85 of an amphipathic helix that mediates membrane association. Neither conclusion is supported directly by high-resolution structural data, but the combined evidence is strong and the interpretations are convincing. The end result is a picture of how TRAPP^{III} associates with membranes and with Rab1/Ypt1 to catalyze nucleotide exchange.

Overall this is an impressive, thorough, interesting study that significantly advances our understanding of the centrally important TRAPP^{III} complex. I have only a couple of minor comments:

1. Although the quantification in Figure 1E looks persuasive at first glance, the primary data in Figure 1D are much less compelling. All four panels in Figure 1D look similar. The issue seems to be that the signal is present over a high background, explaining why the statistical significance of the effect is borderline. Yet the text reads: "Consistent with our hypothesis, the intact TRAPP^{III} complex exhibited robust binding to synthetic liposomes, and the Trs85 subunit was required for stable membrane association (Figure 1E)." This statement needs to be qualified to acknowledge that the method is not actually so robust.

2. In Figure 4, what does "EV" stand for? I assume that "WCE" means whole-cell extract, but that's just a guess. In general, the authors should be careful to define the terms in their figures.

We offer our gratitude to the reviewers and to the editor for their support of our work and for their constructive criticism. We have revised the text based on the concerns presented below and believe that our manuscript has been strengthened accordingly. Please find our point-by-point responses in bold below.

Referee #1:

The authors present a well-written, comprehensive, and multidisciplinary analysis of the TRAPPIII complex (a GEF) and its interactions with Rab1/Ypt1 (a key regulatory GTPase in the secretory and autophagic pathways) and with membranes. While the linchpin of the story is the cryoEM structure of a TRAPPIII-Ypt1 complex (a noteworthy achievement in its own right), the impact of this structure is enhanced by an expansive array of complementary biochemical and in vivo experiments. Together, these reveal the binding site for the key juxtamembrane 'hypervariable' region of the Rab which, in turn, suggests the orientation of the entire TRAPPIII complex on membranes. This hypothesis leads the authors to propose, and then experimentally confirm, that a disordered region of the TRAPPIII-specific subunit Trs85 forms a membrane-inserting amphipathic alpha-helix. The Trs85 amphipathic helix, by helping anchor the TRAPPIII complex to membranes in the proposed orientation, plays an essential role in Rab1 activation.

Taken together, these findings represent a major advance in our understanding of Rab1 biology. The story is mechanistically interesting and biologically important. Overall, while there are always further experiments one could propose, I think the manuscript in its current form is suitable in completeness and impact to merit publication without substantial revision.

Thank you for your thorough review of our work and for your enthusiastic support.

Referee #2:

Using a cryo-EM based approach the authors report the structure of the yeast TRAPPIII complex bound to its cognate GTPase Ypt1. TRAPPIII (is one of two yeast TRAPP complexes - the other being TRAPP II) functions on at least three distinct membrane populations (COPII vesicles, the Golgi, and on autophagosomal membranes). The TRAPPs function as Ypt / Rab GEFs and share the same "core" components whilst containing compositionally distinct subunits that presumably mediate Ypt/Rab-specific interactions as well as membrane associations. In the case of TRAPPIII the distinct component is termed Trs85p (in yeast).

This is detailed and compressive study in which not only novel structural insight is provided, but also importantly, functional insight in cells. Through their structural studies the authors identify how Ypt1 associates with TRAPPIII and in so doing define the

orientation in which TRAPPIII binds to membranes. In addition, the authors also establish that an evolutionarily conserved amphipathic helix in Trs85 is instrumental in mediated TRAPPIII (and Trs85) association with anionic lipid containing liposomes and membranes in cells

I am not an expert in structural studies but nevertheless I would expect this study to be of significant interest to cell biologists in general, and to those who studying membrane trafficking mechanisms in particular (including autophagy).

The manuscript is well written and laid out in a manner which guides the reader.

Thank you for your detailed review and for supporting our manuscript for publication.

Minor points to be addressed:

Figure 4 Panel B - the protein(s) migrating between 75 and 95kDa are labeled as non-specific but are of a similar MW to Trs85 and the Trs85 variant shown in panels A and C (respectively). So, presumably these non-specific proteins are also present in panels A and C. This should be mentioned in the main text (or expanded upon in the corresponding figure legend) to avoid confusion when future readers are scrutinizing the figure.

These non-specific proteins are E. coli contaminants that become more prominent in purifications with lower expression/yield, as is the case for this construct. If they are present in the other constructs, they are present at much lower levels.

To address this point, we have added another sentence (underlined below) to the legend of Figure 4, panel B. It now reads: ***"B. As in A, using a Trs85 mutant containing only the final 198 amino acids. NOTE: Trs85[501-698] is ~25kD in size and now migrates at the same size as some of the core subunits. The species migrating near 85kD are contaminants present due to the lower expression level of this mutant construct."***

Page 9. Would the authors please describe more completely (in genetic terms) what they mean by "sensitized background" with reference to Bet3.

Thank you for this suggestion. For clarity, we have modified the text to read as follows: *"Because loss of Trs85 is synthetically lethal with C-terminal tagging of Bet3 (Sacher et al., 2001), we created a sensitized strain background for trs85Δ complementation tests in which the core Bet3 subunit is tagged at its C-terminus."* The genotype of the strain is also listed in the strain table.

Although it is becoming increasingly common in the literature to see references to amino acid substitutions as mutated amino acids - this does not (in this reviewer's opinion)

make sense - but is rather lab parlance / jargon. The authors should be encouraged to make changes to the main text to rectify this. For example, on the top of page 14.

We agree with this sentiment and where appropriate we have modified all instances of “mutation” to “substitution” or “truncation” and from “mutant” to “substitution mutants”.

Referee #3:

This paper takes a cryo-EM structural approach to examine yeast TRAPPIII, the GEF that activates the Rab1 homolog Ypt1 to regulate the early secretory pathway as well as autophagy. The structure reveals the binding site for the Rab1/Ypt1 hypervariable domain. Trs85 is the subunit that defines the yeast TRAPPIII version of the TRAPP complex, and the authors report that Trs85 apparently has a conserved amphipathic helix that anchors TRAPPIII in the membrane. This work extends previous structural and functional analysis of the core TRAPP complex, and it sheds light on mutations that compromise the function of TRAPPIII.

At 3.7 Å, the resolution of this structure is modest. The authors explain how the orientation bias of the complex precluded a higher resolution analysis, and how they used tilted grids to partially overcome this limitation. The core subunit structures are known from crystallography, while the Trs85 subunit structure was predicted with fairly high confidence and shows a reasonable fit to the cryo-EM data. Cross-linking mass spectrometry was used to validate the structure, so overall the results seem to be solid.

The authors push the system to argue for a specific interaction of the Rab1/Ypt1 hypervariable domain with the TRAPPIII complex, and for the presence in Trs85 of an amphipathic helix that mediates membrane association. Neither conclusion is supported directly by high-resolution structural data, but the combined evidence is strong and the interpretations are convincing. The end result is a picture of how TRAPPIII associates with membranes and with Rab1/Ypt1 to catalyze nucleotide exchange.

Overall this is an impressive, thorough, interesting study that significantly advances our understanding of the centrally important TRAPPIII complex. I have only a couple of minor comments:

Thank you for the positive comments. We appreciate your critiques and support of the manuscript.

1. Although the quantification in Figure 1E looks persuasive at first glance, the primary data in Figure 1D are much less compelling. All four panels in Figure 1D look similar. The issue seems to be that the signal is present over a high background, explaining why the statistical significance of the effect is borderline. Yet the text reads: "Consistent with our hypothesis, the intact TRAPPIII complex exhibited robust binding to synthetic liposomes, and the Trs85 subunit was required for stable membrane association (Figure

1E)." This statement needs to be qualified to acknowledge that the method is not actually so robust.

While we agree that there is some background level of pelleting (both core TRAPP and the TRAPPIII complex do pellet somewhat in the absence of membranes), these experiments have been repeated dozens of times in various forms and the result is always the same (and similar results are presented in Figure 6).

We are puzzled by the reviewer's statement that all the panels in Figure 1D look the same. Panels 3 and 4 in Figure 1D look quite different: in panel 3, most of the complex is in the supernatant whereas in panel 4 most of the complex is in the pellet.

Nevertheless, we have modified the text to be less assertive by removing the word "robust" from the description. We also removed the word "strongly" from the corresponding figure legend.

2. In Figure 4, what does "EV" stand for? I assume that "WCE" means whole-cell extract, but that's just a guess. In general, the authors should be careful to define the terms in their figures.

Thank you for catching those omissions. We have included definitions for the abbreviations within the corresponding figure legends.

1st Revision - Editorial Decision

23rd Mar 2021

Thank you for submitting your revised study. I have now checked your manuscript and the point-by-point rebuttal letter and find that the referees' points have been sufficiently addressed.

However, there are few editorial issues concerning the text and the figures that I need you to address before we can officially accept your manuscript.

2nd Authors' Response to Reviewers

7th Apr 2021

Thank you for submitting your revised study. I have now checked your manuscript and the point-by-point rebuttal letter, and find that the referees' points have been sufficiently addressed.

However, there are few editorial issues concerning the text and the figures that I need you to address before we can officially accept your manuscript.

We are grateful for the positive news and we have made the requested changes. See below for the details, our responses in bold.

2nd Revision - Editorial Decision

11th Apr 2021

I am pleased to inform you that your manuscript has been accepted for publication in The EMBO Journal.

Corresponding Author Name: J. Chris Fromme

Journal Submitted to: The EMBO Journal

Manuscript Number: EMBOJ-2020-107607